# The control and training of single motor units in isometric tasks are constrained by a common input signal

Mario Bräcklein[1], Deren Yusuf Barsakcioglu[1], Jaime Ibáñez[1,2,3], Jonathan Eden[1], Etienne Burdet[1], Carsten Mehring[4,5], Dario Farina[1]*

[1]Department of Bioengineering, Imperial College London, London, United Kingdom; [2]Department of Clinical and Movement Disorders, Institute of Neurology, University College London, London, United Kingdom; [3]BSICoS, IIS Aragón, Universidad de Zaragoza, Zaragoza, Spain; [4]Bernstein Center Freiburg, University of Freiburg, Freiburg im Breisgau, Germany; [5]Faculty of Biology, University of Freiburg, Freiburg im Breisgau, Germany

*For correspondence: d.farina@imperial.ac.uk

Competing interest: The authors declare that no competing interests exist.

**Abstract** Recent developments in neural interfaces enable the real-time and non-invasive tracking of motor neuron spiking activity. Such novel interfaces could provide a promising basis for human motor augmentation by extracting potentially high-dimensional control signals directly from the human nervous system. However, it is unclear how flexibly humans can control the activity of individual motor neurons to effectively increase the number of degrees of freedom available to coordinate multiple effectors simultaneously. Here, we provided human subjects (N = 7) with real-time feedback on the discharge patterns of pairs of motor units (MUs) innervating a single muscle (tibialis anterior) and encouraged them to independently control the MUs by tracking targets in a 2D space. Subjects learned control strategies to achieve the target-tracking task for various combinations of MUs. These strategies rarely corresponded to a volitional control of independent input signals to individual MUs during the onset of neural activity. Conversely, MU activation was consistent with a common input to the MU pair, while individual activation of the MUs in the pair was predominantly achieved by alterations in de-recruitment order that could be explained by history-dependent changes in motor neuron excitability. These results suggest that flexible MU recruitment based on independent synaptic inputs to single MUs is unlikely, although de-recruitment might reflect varying inputs or modulations in the neuron's intrinsic excitability.

## Editor's evaluation

The present study indicates that humans cannot easily learn to control multiple motor units innervating a single muscle independently. These results suggest that common drive to motor units and the size-recruitment principle impose strong constraints on the motor system and, as such, on the use of high-resolution muscle recordings as a means of controlling brain-machine interfaces.

## Introduction

Decoding single motor unit (MU) spiking activity, that is, action potentials discharged by motor neurons and their innervated muscle fibres, non-invasively from the surface electromyogram (EMG), represents a viable alternative to invasive brain recordings for neural human–machine interfaces (*Holobar and Farina, 2021*; *Formento et al., 2021*; *Farina and Holobar, 2015*). One potential application for such non-invasive neural interfaces is to augment the number of degrees of freedom a person can control

by exploiting the fact that hundreds to thousands of motor neurons innervate muscles (*Heckman and Enoka, 2012*). If single MUs could be individually controlled, they could potentially be activated in a multitude of ways, leading to an enormous potential for additional information transfer. New MU activation patterns made of independent control of units could then be used to provide the basis for augmented control signals without impeding the original function of the innervated muscles for natural limb control (*Eden et al., 2022*).

The possibility of controlling part of the MUs in a muscle independently would indeed provide a separation from the natural control of force, mainly provided by the population behaviour, and a vast augmentation resource from independent control. Nonetheless, while being a very attractive potential mechanism for artificial augmentation, the independent control of MUs would imply an increased computational load by the central nervous system (CNS) without any known *natural* functional benefit for the human motor system. Indeed, most previous observations indicate that, contrary to independent control, single MUs tend to be activated in a very stereotyped way which is determined by the common input received by functional groups of MUs (*Negro et al., 2016b*; *Tanzarella et al., 2021*; *Farina et al., 2014*) and by their biophysical properties (i.e. *Henneman's size principle* stating a dependency between the exerted force at which an MU starts to contribute and the neuron's size; *Henneman, 1957*; *Henneman et al., 1974*). Since the size of the soma is inversely related to the membrane resistance, smaller motor neurons discharge action potentials earlier and at higher rates than larger neurons for the same net excitatory synaptic input (*Stein et al., 2005*). For this reason, if a pool of MUs receives the same common input, MU recruitment is solely dependent on the MU anatomy and on the intrinsic excitability of the motor neurons. The size principle has been observed in several muscles (*Desmedt and Godaux, 1977a*; *Oya et al., 2009*; *van Zuylen et al., 1988*; *Thomas et al., 1987*; *Thomas et al., 1986*; *Monster and Chan, 1977*) and appears to remain robust in various scenarios (*Thomas et al., 1987*; *Thomas et al., 1986*; *Adam and De Luca, 2003*; *Fling et al., 2009*; *Desmedt and Godaux, 1977b*; *Thomas et al., 1978*; *Jones et al., 1994*).

Previous works have tried to challenge the perspective of single MUs only being activated in a predetermined fashion (*Formento et al., 2021*; *Basmajian, 1963*). A recent study even provided evidence that indeed there is, to some degree, a neural substrate that would allow for the selective cortical control of MUs via descending pathways (*Marshall et al., 2021*). However, so far, it is unclear whether humans can learn to leverage such a potential neural structure for selectively activating MUs by converting the common neural input received by an MU pool to independent inputs to individual MUs and thus change the original MU recruitment.

This study examined whether humans could control pairs of MUs innervating the same muscle flexibly. Further, it addressed how this potential ability depends on the similarity of the MU pairs in size or, equivalently, in recruitment threshold. For this purpose, we used a neural interface that provided subjects with biofeedback on the activity of individual MUs (*Barsakcioglu et al., 2021*). Subjects were encouraged to navigate a cursor inside a 2D space into different targets as quickly as possible by selectively recruiting different MUs using feedback received on both MU activity and resulting cursor movement. This allowed us to assess if subjects were able to leverage potential selective descending pathways that would facilitate independent synaptic input to individual MUs or if, instead, they used control strategies based on a common input to the MU pair. After several days of training, all subjects achieved the target-tracking task. However, the control strategies used that allowed individual MU activation did not leverage potential selective inputs to single MUs. Instead, subjects strongly favoured control strategies based on a common input signal combined with changes in intrinsic motor neuron excitability due to history-dependent physiological properties of the activated MUs.

## Results

### Target task

During the main experimental sessions, on average, 11.04 ± 3.34 MUs were reliably decomposed (online surface EMG decomposition) per subject (N = 7). An identified MU pool from a subject sorted based on recruitment order is shown in *Figure 1A*. As indicated by the green and red marks, the order in which MUs are de-recruited often differed from the recruitment order. For example, once recruited, an MU could keep discharging action potentials even when the exerted force level was below the initial recruitment threshold. For the target task (see Materials and methods), two pairs of

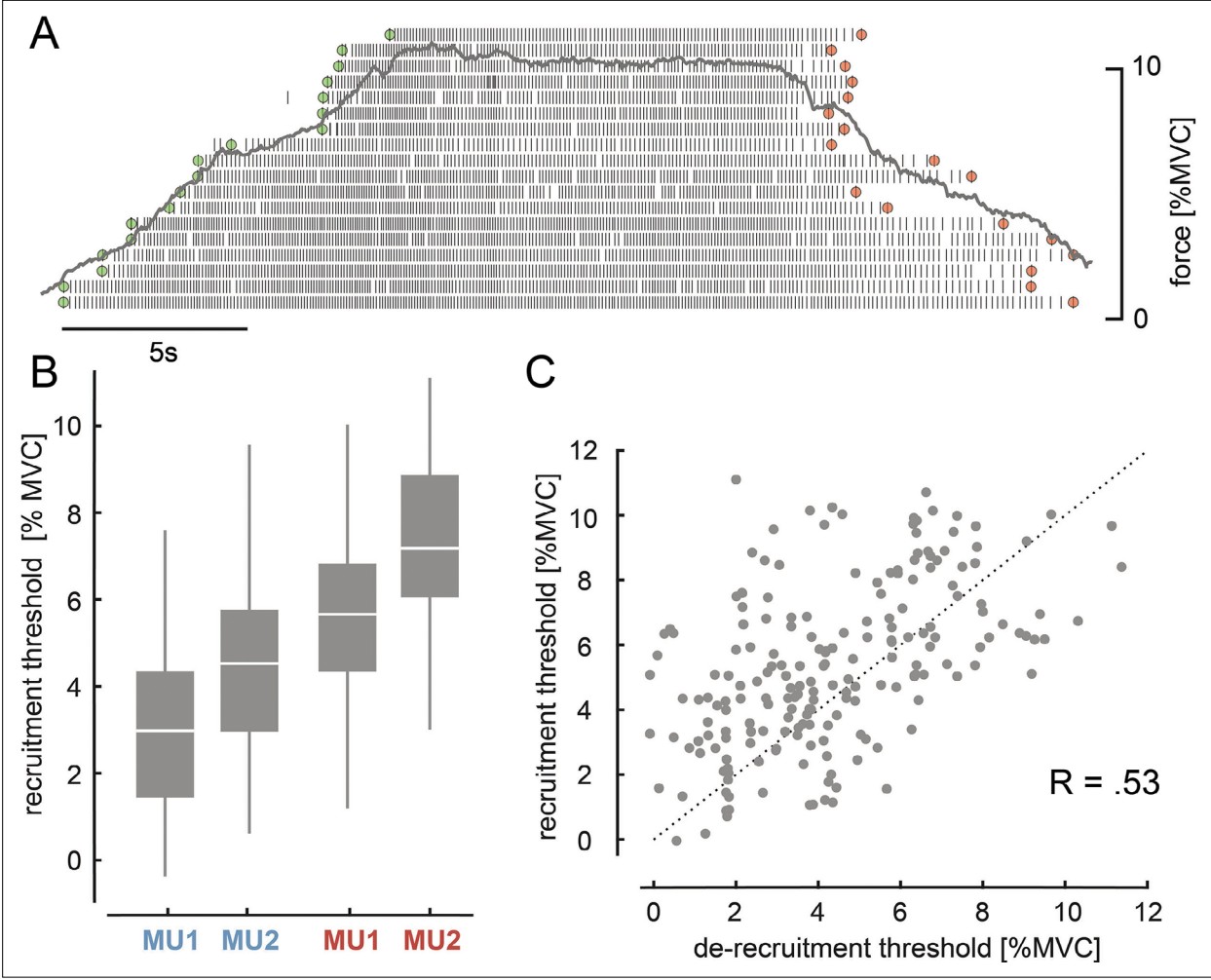

**Figure 1.** Motor unit (MU) recruitment and de-recruitment. (**A**) The identified MU pool ranked on the recruitment order of one representative subject is shown with the underlying force profile (grey). Time points of recruitment (green) and de-recruitment (red) for each MU are marked. (**B**) Recruitment levels for MU1 and MU2 of the lower (blue) and higher threshold pair (red) across all subjects and days are shown with their median and quartiles. (**C**) Recruitment and de-recruitment threshold for the selected MUs across all days showed a significant relationship (p<0.001). Dashed line indicates the diagonal. The three subjects for whom no de-recruitment thresholds were determined were neglected in this correlation analysis.

The online version of this article includes the following figure supplement(s) for figure 1:

**Figure supplement 1.** High-density surface electromyogram (HDsEMG) from a single subject.

MUs with a small difference in recruitment threshold were selected out of the entire pool. Each pair was selected either from the first or from the last recruited half of the MU pool. *Figure 1B* visualises the recruitment thresholds of these selected MUs. Within pairs, MU1 was recruited before MU2 and the lower threshold pair (blue) was recruited before the higher threshold one (red). Recruitment and de-recruitment threshold of the selected MUs showed a strong relationship across days and subjects (Spearman's correlation coefficient, R = 0.53, p<0.001; see *Figure 1C*). On average, the de-recruitment threshold was –1.11 ± 2.44% maximum voluntary contraction (MVC) smaller than the recruitment one, and in 66.7% a selected MU was de-recruited at a force level below its initial recruitment threshold. The overall observed effect, however, was weak (see Materials and methods, p=0.111).

During the target task, subjects were asked to navigate a cursor inside a 2D space by modulating the discharge rate of MU1 and MU2. Three different targets inside the 2D plane were used to encourage subjects to activate both MUs independently despite their different position within the recruitment order. An entire trial cycle of this target task is illustrated in *Figure 2*.

For example, to reach TIII, subjects must keep the higher threshold unit MU2 active while keeping the lower threshold unit MU1 off. *Figure 3A* shows, as an example, the average cursor position during

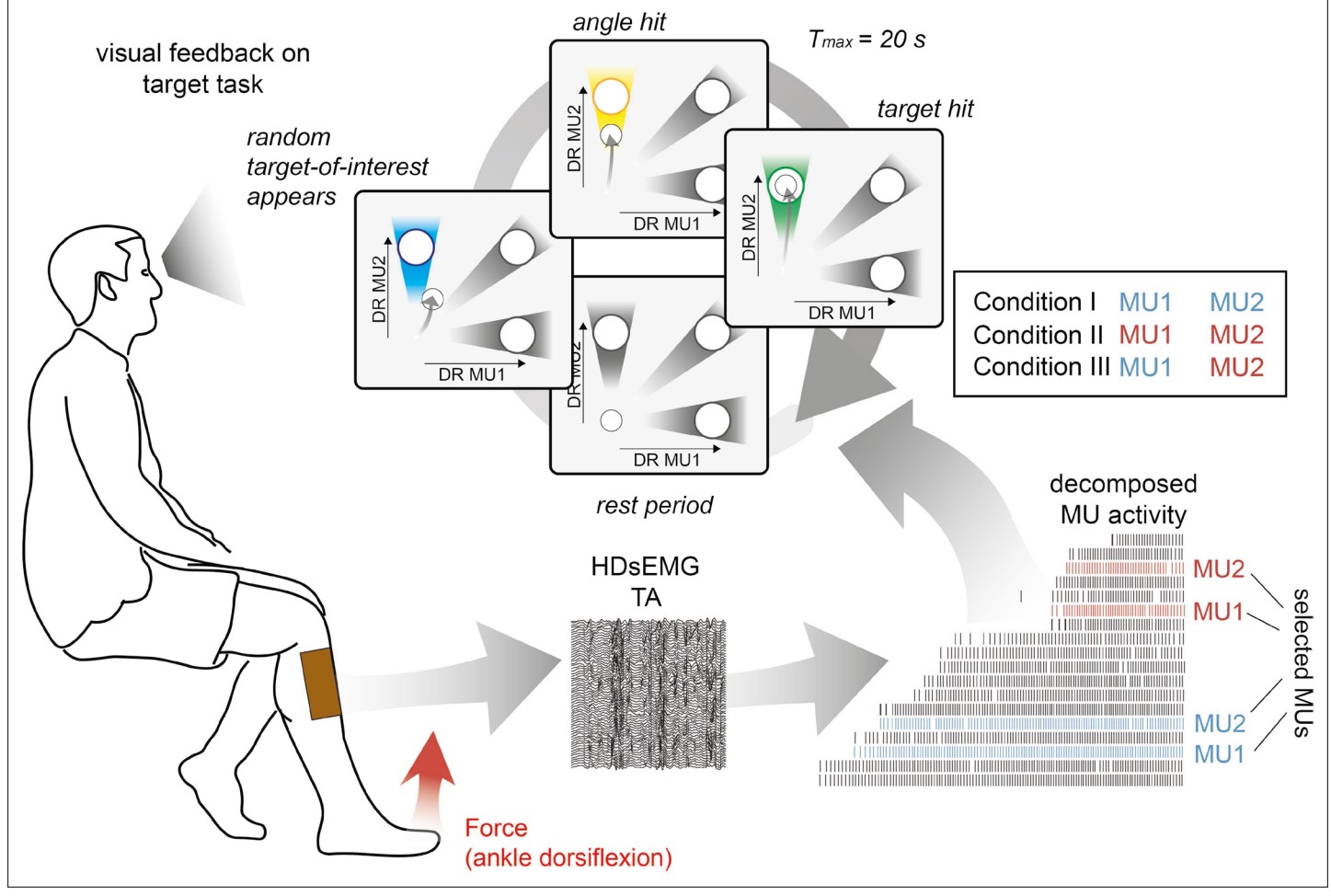

**Figure 2.** Schematic overview of the target task. High-density surface electromyogram (HDsEMG) of tibialis anterior muscle (TA) was acquired and decomposed from the underlying neural activity in real time. Concurrently, the force due to dorsiflexion of the ankle (red arrow) and bipolar electromyogram (EMG) of fibularis longus (FL), lateral head of the gastrocnemius muscle (GL), and medial head of the gastrocnemius muscle (GM) were recorded. The identified motor unit (MU) pool was ranked accordingly to the recruitment order. Two pairs of MUs with a similar recruitment threshold were selected from the initial (blue) and the latter recruited half (red). During the target task, subjects were instructed to navigate a cursor inside a 2D space by modulating the normalised discharge rate (DR) of MU1 and MU2. The selection of MU1 and MU2 was determined by three different conditions. In condition I, MU1 and MU2 were coming from the low recruitment threshold pair (blue), in condition II from the high recruitment threshold pair (red), while in condition III, the lowest threshold MU of the low threshold pair was pooled with the highest threshold MU of the high threshold pair. During the target task, subjects were asked to stay inside the origin until the target-of-interest (blue) appeared (randomly selected). By navigating the cursor inside the angle area around the target-of-interest, subjects were granted an angle hit (yellow). The trial was terminated when either the subject managed to place and hold the cursor inside the target area (target hit, green) or more than 20 s had passed since the target-of-interest appeared. In each condition, 30 targets are shown, that is, each target 10 times.

target hits and nearest misses for each target-of-interest across conditions towards the beginning and end of training for a single subject. At the beginning of the task, the subject failed in the majority of trials to place and hold the cursor inside the target-of-interest. With training, the ability to place the cursor inside the designated target area improved. As shown in *Figure 3A*, TH was hit in all 30 trials, and in only 4 trials, the subject could not hit TI. Moreover, the nearest misses for TIII in the 12th training session were closer to the target centre than on day 1. This improvement in hitting targets and angles over several training days was observed across all subjects (see *Figure 3B*). The target hit rate improved from the first to the last day of training from 41.19 ± 17.76% to 67.04 ± 18.17% (two-sided Wilcoxon signed-rank test, p=0.028). A similar trend was observed for the angle hits with an improvement from 64.37 ± 13.15% to 81.11 ± 9.84% (two-sided Wilcoxon signed-rank test, p=0.016). According to the distanced-based performance metric (defined in Materials and methods), the performance per subject across targets and conditions improved from the first to the last day of training

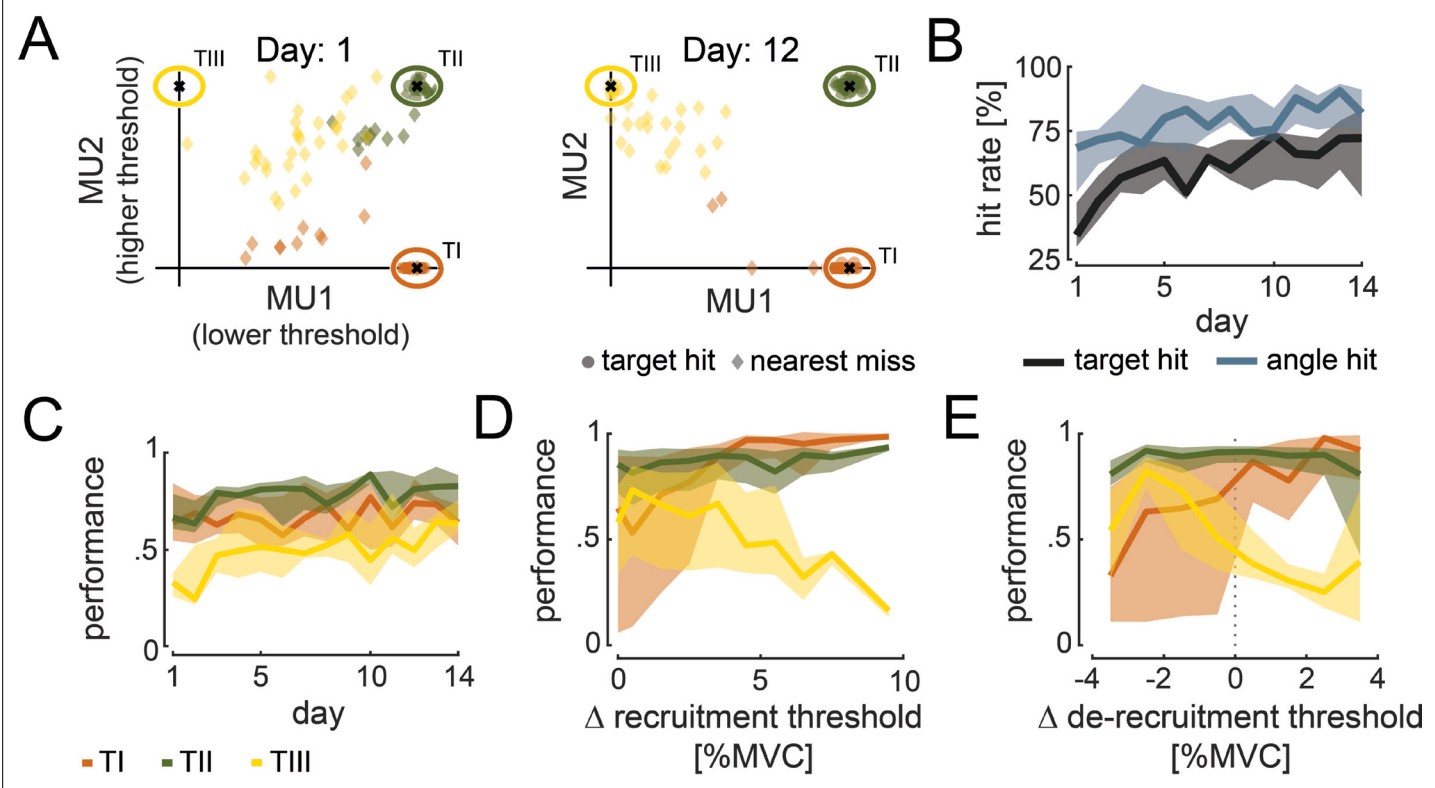

**Figure 3.** Cursor movement and performance during target task. (**A**) Average cursor position during target hits (circle) and nearest misses (diamond) across conditions is shown for the 1st and 12th day of experiments for TI (orange), TII (green), and TIII (yellow) of one subject. (**B**) Target (black) and angle hit rate (blue) across subjects, conditions, and targets-of-interest are shown with their medians (solid line) and 25 and 75% quartiles (shaded areas) across days. (**C**) Performance values across subjects and conditions for TI (orange), TII (green), and TIII (yellow) are shown with their medians (solid line) and 25 and 75% quartiles (shaded areas) across days. (**D, E**) Performance values corresponding to the difference in recruitment threshold (**D**) and de-recruitment threshold (**E**) between motor unit (MU)2 and MU1 are shown across subjects and conditions for the last 5 days of training. The median (solid line) and 25 and 75% quartiles (shaded area) for TI (orange), TII (green), and TIII (yellow) are illustrated in steps of 1% maximum voluntary contraction (MVC).

The online version of this article includes the following figure supplement(s) for figure 3:

**Figure supplement 1.** Additional metrics concerning the target-tracking task.

from 0.56 ± 0.09 to 0.69 ± 0.10 (two-sided Wilcoxon signed-rank test, p=0.016). Despite this clear improvement in performance across days, the ability to move the cursor towards the target-of-interest enhanced differently across targets (see *Figure 3C*). The performance in hitting TI did not significantly improve from the first day of training to the last day (from 0.67 ± 0.18 to 0.68 ± 0.11; two-sided Wilcoxon singed-rank test, p=0.938). Similarly, no significant improvement was observed for TII (from 0.69 ± 0.11 to 0.80 ± 0.13; two-sided Wilcoxon signed-rank test, p=0.109). However, a significant improvement in performance was detected when subjects were asked to move towards TIII (from 0.33 ± 0.12 to 0.59 ± 0.17; two-sided Wilcoxon signed-rank test, p=0.016). Furthermore, across conditions and days, we only found minor within-session learning (mean correlation between performance and consecutive trials across days and condition for TI: 0.037 ± 0.078, TII: 0.066 ± 0.094, TIII: 0.043 ± 0.074, see *Figure 3—figure supplement 1C*) with no significant difference across targets (Friedman, p=0.103).

Taken together, target and angle hits, as well as the performance metric, indicate that subjects improved in navigating the cursor towards the target-of-interest across days. However, the main improvement was observed for reaching TIII, which started from poor initial performance. Moreover, subjects experienced a steep learning curve at the beginning of the experiment, while the learning rate seemed to decrease towards the end. For this reason, further analysis only focuses on the last 5 days of training to avoid additionally induced variability by greater learning rates at the beginning of training.

The performance of reaching each target did not solely depend on the target's position but also on the difference in recruitment and de-recruitment threshold within the selected MU pair, that is, the force difference between the onset and offset of activity between MU2 and MU1 measured during the initial force task (see Materials and methods). *Figure 3D and E* illustrate the performance over the within-pair difference in recruitment and de-recruitment threshold, respectively, for each target-of-interest. Performance per subject in reaching TI significantly increased from selected MU pairs with a small difference in recruitment threshold (0–1% MVC) to those with a high difference (3–10% MVC) from 0.61 ± 0.16 to 0.90 ± 0.05 (two-sided Wilcoxon signed-rank test, p=0.016). To reach TI, only MU1 must be active. Therefore, this result indicates that subjects performed better in keeping only MU1 active while not activating MU2 when the difference in their recruitment threshold was high. This may be due to less accuracy needed in the force generated when the recruitment threshold difference is large. For example, if MU1 gets recruited at 2% MVC while MU2 at 5% MVC, the subject could potentially exert any force between 2 and 5% MVC to keep only MU1 active without MU2 to ultimately hit TI. A more precise force level needs to be generated when this difference is smaller. For TII, no dependency in performance and the within-pair difference in recruitment threshold was detected (from 0.79 ± 0.07 to 0.72 ± 0.08, two-sided Wilcoxon signed-rank test, p=0.578). On the contrary, the performance in reaching TIII decreased significantly for larger differences in recruitment threshold within the selected MU pair from 0.66 ± 0.09 to 0.55 ± 0.04 (two-sided Wilcoxon signed-rank test, p=0.016). This indicates that subjects experienced difficulties in keeping MU2 active while MU1 was inactive in order to reach TIII when their difference in recruitment threshold was relatively large (3–10% MVC). The difference in recruitment threshold between MU1 and MU2 correlated with the difference in de-recruitment threshold (Spearman's correlation coefficient, $R$=0.46, p=0.001; see *Figure 3—figure supplement 1A*). Thus, similar implications on the relationship between performance in reaching the targets and the difference in de-recruitment threshold between MU1 and MU2 can be drawn (see *Figure 3E*). Subjects became better in reaching TI for a positive difference in de-recruitment thresholds (i.e. MU2 is de-recruited before MU1; 0.76 ± 0.05) than for negative ones (i.e. MU2 is de-recruited after MU1;. 044 ± 0.11; all four subjects for whom the de-recruitment threshold was recorded increased in their performance from negative to positive). When subjects had difficulties in activating MU1 without activating MU2 to reach TI (e.g. when the recruitment threshold of both MUs is similar, see above), they usually tried to switch off MU2 while keeping MU1 active to reach again towards TI. However, if MU2 was de-recruited after MU1, the subjects needed to move back to the coordinate origin to switch off MU2 before they could start a new attempt in reaching towards TI. This required additional time, resulting in lower performance. However, when MU2 was de-recruited before MU1, the subjects only needed to switch off MU2 (while MU1 remained active) before reaching TI again. For TII, no dependency between difference in de-recruitment threshold and performance was observed (from 0.83 ± 0.05 to 0.83 ± 0.04; the performance never varied more than 0.02 from a negative to a positive difference in de-recruitment threshold per subject). When asked to reach TIII, subjects decreased in performance from 0.57 ± 0.04 to 0.40 ± 0.02 for negative and positive differences in the de-recruitment threshold, respectively. The better performance for negative differences in de-recruitment thresholds may be explained by a possible strategy to reach TIII by leveraging the lower de-recruitment threshold of MU2 compared to MU1. Because of this, subjects could activate both MUs first before lowering the force level to switch off MU1 while MU2 remained active due to its lower de-recruitment threshold. This would result in cursor movement towards TIII.

During the last 5 days of training, the majority subjects reported that they felt having control over MU2 when reaching towards TIII (14% of cases subjects indicated having no control over MU2; see *Figure 3—figure supplement 1D*). Moreover, they declared the usage of a clear strategy to establish such control, which varied in cognitive demand across subjects and days. In 95% of all cases, this control strategy was described as a rapid increase in force due to dorsiflexion of the ankle followed by a slow release until the cursor moves towards the vertical axis.

All subjects improved their performance during training. To better understand which strategies have emerged, ultimately enabling subjects to recruit and de-recruit single MUs, we analysed the cursor trajectories during the task. The cursor movement for one subject during the last day of training (condition I) for all 30 trials is visualised in *Figure 4*. While the subject was able to hit all targets before the trials ended, the cursor trajectories did not always mimic the straight path, to the target centre. When asked to move towards TI, the subject moved the cursor along the horizontal axis. In trials 1, 2,

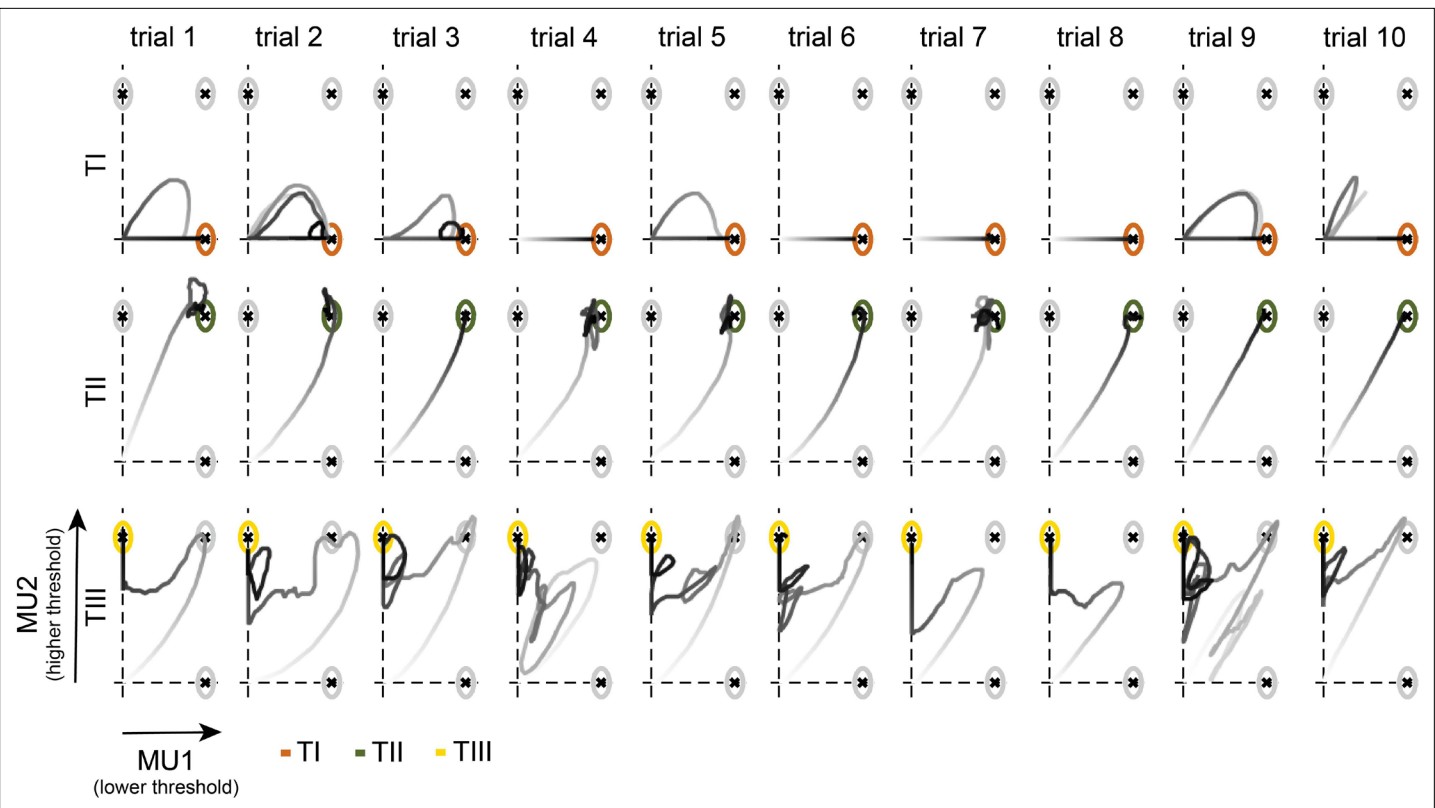

**Figure 4.** Cursor trajectories of one subject during the target task. Cursor movement towards TI (orange, top), TII (green, centre), and TIII (yellow, bottom) in each trial for condition I on the 14th day of one representative subject is shown. Trials 1–10 indicate the 1st to the 10th appearance of each target-of-interest. The grey intensity of the cursor trajectories increases over time within the trial.

3, 5, 9, and 10, the subject could not hit TI in the first attempt but returned to the origin to then move towards the target centre directly. For TII, in all cases, the subject moved directly along the diagonal towards the target-of-interest. For TIII, however, instead of moving directly towards the target centre, the subject moved the cursor towards TII first and then towards the vertical axis to finally hit TIII. This observation is in line with the descriptions provided by the questionnaire (see *Figure 3E*), that is, increase in force to activate both MUs, followed by a decrease in the force until MU1 switches off, and ultimately adjusting the force level to move along the vertical axis towards the target centre.

By analysing the unintended target and angle hits (see *Figure 5A*), the probability that the cursor was moved towards unselected targets while trying to hit the target-of-interest was quantified.

While only a few unintended hits occurred across targets and conditions, unintended angle hits of TII happened multiple times when subjects tried to reach for TIII. In fact, across all conditions, almost all subjects conducted most of their unintended hits when aiming for TIII (with unintended hits in TII). Only one subject had a higher rate of unintended hits when aiming for TII (with unintended hits in TI) in condition I. Therefore, the strategy to reach TIII, that is, moving towards TII first, as illustrated in *Figure 4* and interpreted by the questionnaire answers, can be observed across subjects. Moreover, the few unintended hits when reaching towards TI and TII suggest that subjects established control strategies that allowed for a direct movement towards the target-of-interest. These clear control strategies, as well as the difference in learning rate across targets, suggest that subjects were able to activate MU1 alone (to reach TI), MU1 and MU2 together (to reach TII), but could not volitionally activate MU2 before MU1 (TIII). In fact, during the last 5 days of training, 76, 78, and 94% of all successful attempts, that is, at least an angle hit, of going towards TI were achieved without activating MU2 once during the trial while it was only 7, 2, and 1% for TIII (MU2 only without MU1) in conditions I, II, and III, respectively. Such rare and occasional deviation from the normal recruitment of two MUs was already observed in one of the first studies employing single MU biofeedback (*Terjung, 1981*). The percentage of these *direct movements* towards TI and TIII with respect to the difference in recruitment threshold

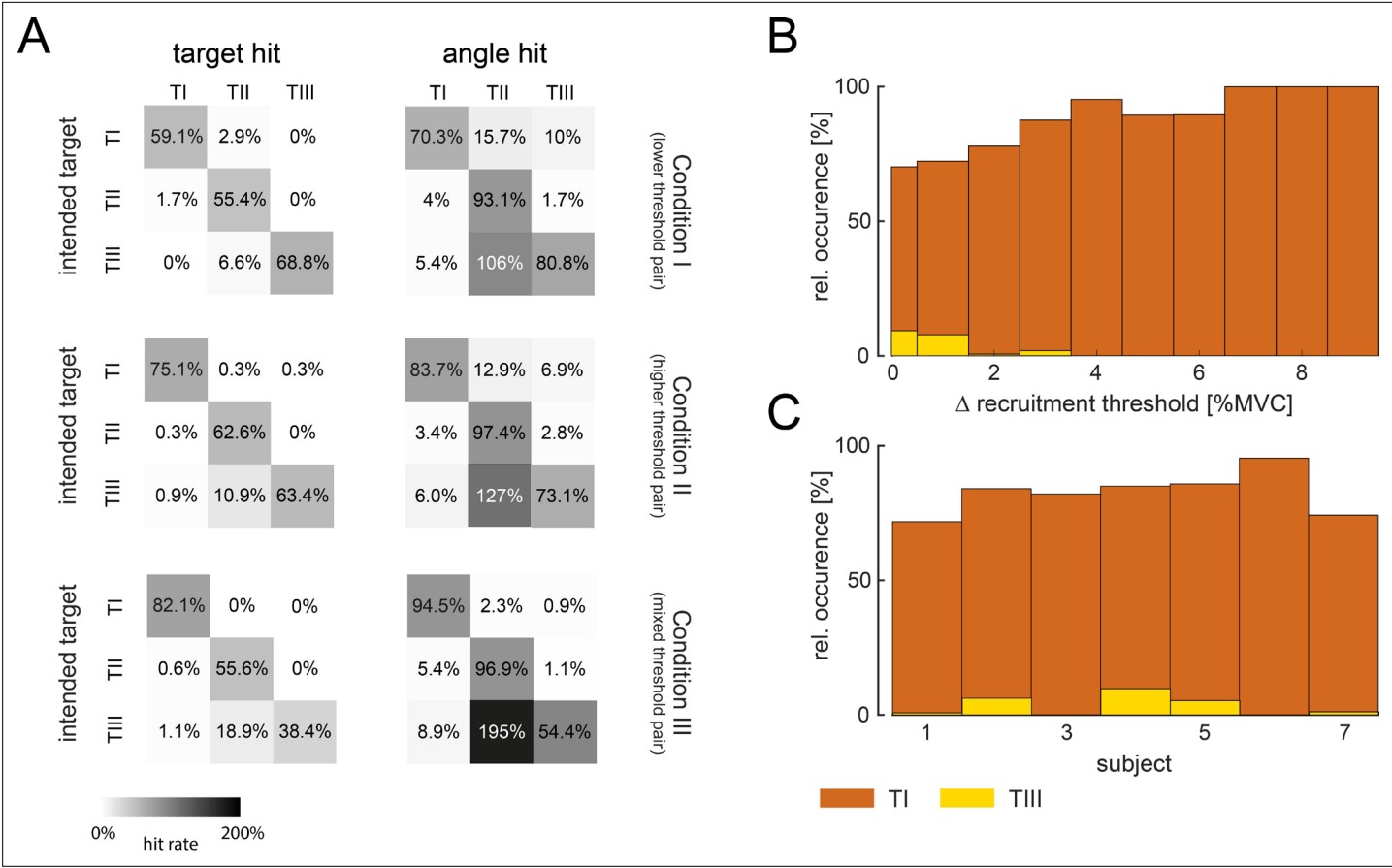

**Figure 5.** Movement towards targets-of-interest. (**A**) Relative hit rates of intended and unintended hits (grey) of targets and angles are shown for the targets-of-interest TI, TII, and TIII during the last 5 days of training across subjects for condition I (top), condition II (centre), and condition III (bottom). Note that the hit rates above 100% can be reached for unintended hits when subjects re-entered the target before the trial ended. Colour intensity corresponds to the hit rate. Relative occurrence of direct movement towards TI (orange), that is, only activating motor unit (MU)1 without MU2, and TIII (yellow), that is, only activating MU2 without MU1, during successful attempts (at least angle hits), is shown with respect to the difference in recruitment threshold between MU2 and MU1 (**B**) and across subjects (**C**).

within the selected pair and subjects is shown in *Figure 5B*. While direct movements towards TI increased with a larger difference in recruitment threshold, direct movements towards TIII were very rare and only possible with MUs recruited at very similar force levels, that is, with small differences in recruitment thresholds that led to variable recruitment orders given sudden excitatory inputs at the beginning of the trials. Also, all subjects moved directly towards TI in more than 70% of all successful attempts. Although only three subjects navigated the cursor directly towards TIII in more than 5% of all successful attempts (see *Figure 5C*). Furthermore, 8.33% of these direct movements towards TIII occurred after TI-instructed trials. In the remaining 91.67% of the trials, direct movements towards TIII occurred after TII- or TIII-instructed trials, which required the activation of MU2 to reach the target. In contrast, direct movements towards TI occurred in 25.93% after TIII-instructed trials or even as the first trial in the session.

In these rare cases in which direct movements towards TIII ended, at least, in an angle hit, the level of force and global EMG of tibialis anterior muscle (TA), fibularis longus (FL), lateral head of the gastrocnemius muscle (GL), and medial head of the gastrocnemius muscle (GM) at the time point of recruitment of MU2 was compared with the corresponding values obtained at normal recruitment of MU2 during the initial force ramps (see Materials and methods). During direct movement towards TIII, MU2 was recruited on average at a 48.43 ± 22.46% lower force level than during ramp recruitment. Also, the global EMG values at recruitment of MU2 were, on average, slightly lower during direct movements towards TIII (TA: –15.91 ± 9.67%; FL: –18.75 ± 23.27%; GL: –9.99 ± 31.53%; GM: –0.02 ± 27.31%).

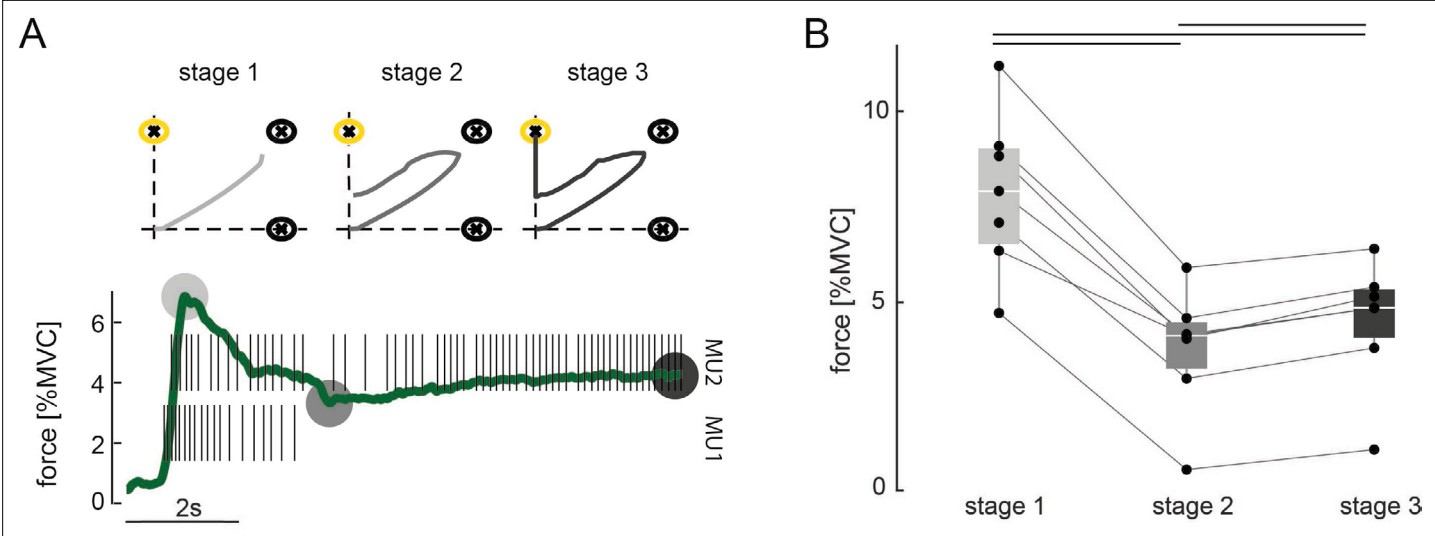

**Figure 6.** Three-stage approach to hit TIII. (**A**) Force due to ankle dorsiflexion (green) and the discharge behaviour of the selected motor unit (MU) pair during a successful attempt of hitting TIII (yellow) for a representative subject. The subject used a three-stage approach to achieve the target task: stage 1: increasing the force to orderly recruit both MUs; stage 2: reducing the force until MU1 stops firing while the cursor is placed along the vertical axis; stage 3: slightly increasing the force again to manoeuvre the cursor inside TIII. Grey circles mark the characteristic force values of each stage. Stage 1: maximum force while both MUs are active; stage 2: minimum force after MU1 stopped firing; stage 3: force during hit of TIII. The corresponding cursor movement for each stage is shown on the top. Grey intensity increases with stages. (**B**) Characteristic forces (due to ankle dorsiflexion) are shown with their median and quartiles at each stage of control for all subjects across all conditions during all TIII hits in the last 5 days of training. Each dot represents a subject, and corresponding values are connected via the lines. Black bars indicate a significant difference with p<0.05.

The online version of this article includes the following figure supplement(s) for figure 6:

**Figure supplement 1.** Various successful trials of hitting TIII for different subjects, conditions, and training days.

However, in the vast majority of cases, target hits of TIII were not achieved by direct movements towards the target centre. Instead, subjects used a three-stage approach to place the cursor inside TIII, as observed in *Figure 4* (e.g. TIII, trial 7). First, subjects navigated the cursor along the diagonal towards TII (stage 1) before in the second stage moving towards the vertical axis. In the third stage, subjects manoeuvred the cursor along the vertical axis inside TIII. The discharge rate of MU1 and MU2 and the exerted force during indirect movement towards TIII are shown for a representative subject in *Figure 6A* (more single-trial examples for TIII-instructed trials are shown in *Figure 6—figure supplement 1*). During the first stage, the subject increased the force to orderly recruit MU1 and MU2. Once both MUs were active, the subject decreased the force level in stage 2 to a minimum so that MU1 stopped firing while keeping MU2 active. In the third stage, the force was slightly increased to match the necessary discharge rate of MU2 to reach TIII without reactivating MU1. To assess whether such force modulation during indirect hits of TIII could be observed across subjects and conditions, characteristic forces (due to ankle dorsiflexion) for each stage were compared in *Figure 6B*. The characteristic force during stage 1 was the mean force in a 100 ms window around the maximum force when both MUs were active, that is, discharge rate greater than 5 pulses per second (pps). In stage 2, the characteristic force was estimated by averaging the force inside a 100 ms window at the minimum force level after switching off MU1 while MU2 continued firing action potentials. In stage 3, the characteristic force was set as the mean force during the hold period preceding a target hit of TIII. Across conditions, all subjects used significantly different force levels during each stage (Friedman, p<0.001, two-sided Wilcoxon signed-rank test Bonferroni-corrected, always p<0.05). During stage 1, the exerted force level was the greatest while being reduced to a minimum in stage 2, before being slightly increased again in stage 3. Furthermore, for the four subjects for whom the de-recruitment threshold was determined, 24.31% of all indirect target hits of TIII were achieved while MU2 was de-recruited before MU1 during the initial force ramps. This indicates that this three-stage approach also worked for pairs of MUs for which the de-recruitment threshold was not reversed to the recruitment one (see *Figure 6—figure supplement 1C and D*).

The force, global TA, FL, GL, and GM EMG values in 100 ms window centred around the onset of MU activity were compared before and after the experiment to investigate potential changes in the recruitment order due to single MU modulation. A subtle decrease in force (–0.18 ± 2.33% MVC, neither did the de-recruitment force change critically: –0.16 ± 2.28% MVC) and global TA EMG (–2.35 ± 22.62%) were identified. However, the global EMG from the lower leg muscles not used for the MU decomposition increased slightly (FL: 5.45 ± 28.58%; GL: 14.12 ± 37.53%; GM: 18.41 ± 26.51%). These changes indicate that the overall recruitment order did not change critically due to the imposed single MU modulation. The increase in activity in the lower leg muscles not directly involved in ankle dorsiflexion relative to the agonist muscle might be explained by induced fatigue towards the end of the experiment (*Patikas et al., 2002*).

## Validation of online EMG decomposition during recruitment/de-recruitment

Surface EMG decomposition with methods similar to those used in this study has been previously extensively validated, including by comparison with intramuscular EMG decomposition (*Barsakcioglu*

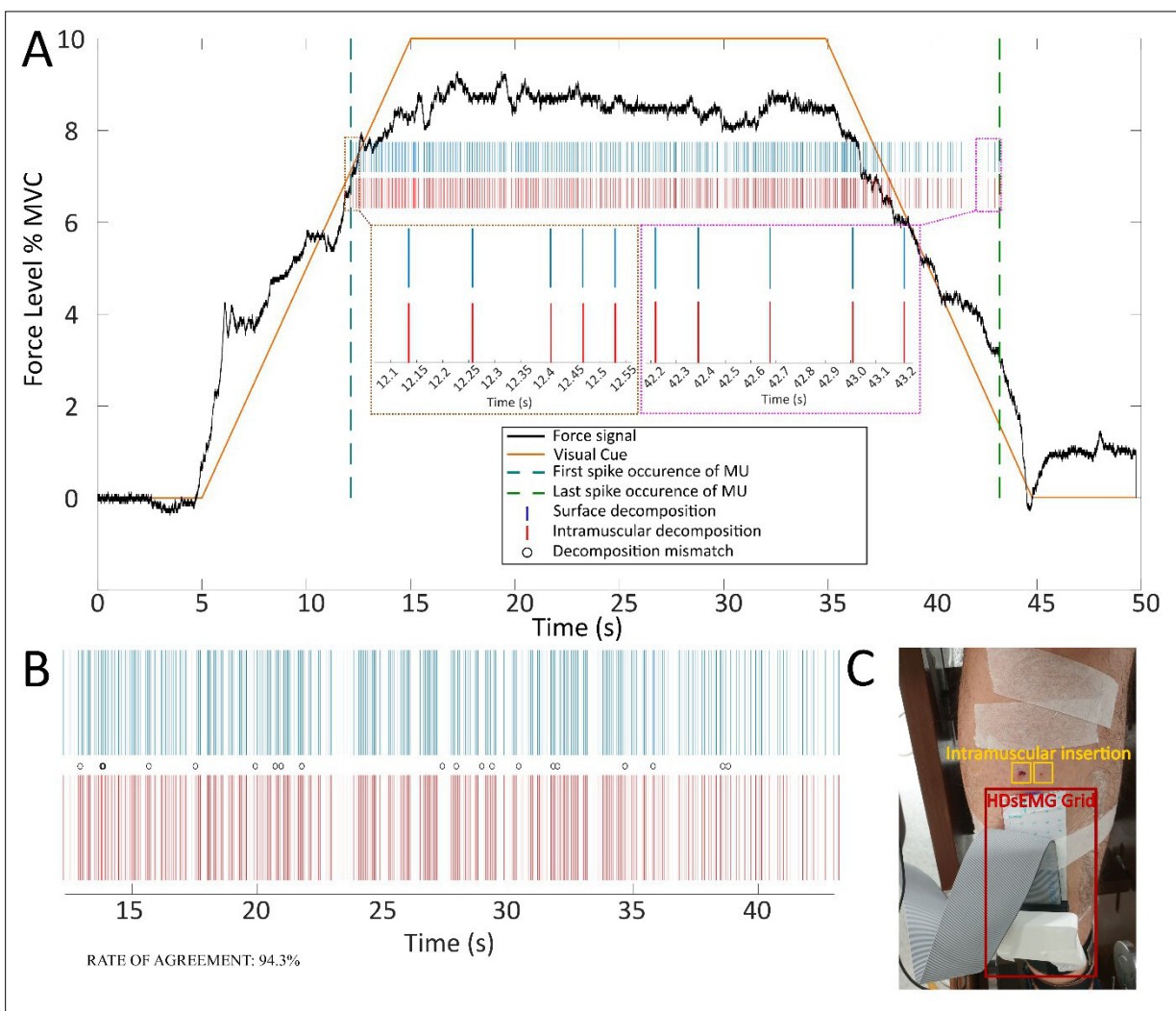

**Figure 7.** An example two-source validation of surface and intramuscular recorded electromyogram (EMG) signals during recruitment/de-recruitment. (**A**) The visual cue provided to the user during the trial (orange solid line), the force feedback on the ankle dorsiflexion (black solid line), the first and last occurrence of motor unit (MU) firing (teal and green dashed lines, respectively) and two subpanels showing the first and last five MU discharges identified for intramuscular and surface decomposed signals (brown and magenta subpanels). (**B**) The decomposed MU spike train from surface EMG (blue) and intramuscular EMG (red), as well as decomposition mismatches, identified (black circles). (**C**) The experimental setup for concurrent intramuscular and surface EMG recordings.

*et al., 2021*). However, this study involved recruitment and de-recruitment of MUs and the accuracy in detecting the recruitment and de-recruitment times was of crucial importance for interpreting the results. Therefore, a validation study was performed in which the onset and offset of MU activity detected from the surface EMG was compared with the corresponding MU activity detected from concurrent intramuscular recordings (see Materials and methods). *Figure 7* shows an example of such a comparison. In this example, one MU was commonly detected by the (online) surface and intramuscular EMG decomposition. The rate of agreement over the full interval of activity of the MU in this representative case was >94%, which is similar to previous validation results obtained in stable-force conditions (*Barsakcioglu et al., 2021*). From *Figure 7*, it can be noted that there were no errors in the first discharges at recruitment and in the last discharges at de-recruitment; therefore, onset and offset of this MU activity were detected highly accurately. In this example, the rate of agreement was also computed separately for the different phases of the contraction (ramp-up, constant force, ramp-down). The ramp-up phase of this MU activity lasted between 12.1 and 15.0 s of the contraction time and in this phase the rate of agreement was 91.7%. The constant-force phase of the contraction for this MU lasted from 15.0 to 35.0 s, during which the rate of agreement was 94.3%, and the ramp-down phase lasted between 35.0 and 43.2 s with an estimated accuracy (rate of agreement) of 96.1%.

The analysis of the other contractions recorded for validation led to results similar to those presented representatively in *Figure 1*. As in previous validation studies using intramuscular EMG as reference, the number of MUs commonly detected by the surface and intramuscular EMG was very small (range 0–1 per contraction). This is due to the different detection volume of the two recording methods, as previously discussed (*Barsakcioglu et al., 2021*), and is in agreement with previous validation studies done at constant force. Out of the 36 contractions during which validation signals were recorded (three subjects, six slopes, two trials per slope), common MUs were detected in only 16 contractions. In all cases in which common MUs were detected, the rate of agreement was >90% and the errors in decomposition were equally distributed in all phases of the contractions (on average, 93.2 ± 1.3%, 92.6 ± 1.5%, and 95.5 ± 2.0% for the three phases of the contraction, respectively), without any preferential distribution around the intervals of recruitment and de-recruitment. Notably, we never observed an error in the first discharge (onset) or the last discharge (offset), meaning that the recruitment and de-recruitment times were detected by the online decomposition with an accuracy of 1 ms (margin used in the time alignment between discharge times detected by the surface and intramuscular decomposition).

From the validation tests, we concluded that the online surface EMG decomposition in variable force contractions with recruitment and de-recruitment achieved similar performance as the decomposition in static contractions.

## Discussion

Volitional and flexible control of single MUs could revolutionise neural-interface applications. Here, we used real-time biofeedback on single MU activity to encourage subjects to learn independent control of pairs of MUs. Our results showed that subjects could gain control over four MUs from a single muscle. The control strategies that emerged, allowing for selective MU control, were limited by the presence of a common input to the MU pool. Therefore, subjects did not exploit potential neural structures with selective inputs to individual MUs.

In this study, the identified MU pools were ranked based on their recruitment order. We have shown that the de-recruitment threshold does not always match the recruitment one. While small deviations between recruitment and de-recruitment threshold may be explained by a slight increase in co-contraction during the declining force ramp (*De Luca and Mambrito, 1987*), larger mismatches might be due to intrinsic neuromodulatory and inhibitory mechanisms. The latter are triggered following the activation of an MU, which can disrupt the simple dependency of recruitment order from neuron size and input received (*Heckman and Enoka, 2012*). One potential mechanism contributing to this neuromodulation in motor neurons is persistent inward current (PIC) (*Binder et al., 2020*). PICs, that is, descending neuromodulation from the brainstem to the spinal cord, can alter the excitability of motor neurons which may lead to a self-sustained firing of action potentials and thus to de-recruitment at a force level lower than the recruitment threshold, that is, recruitment de-recruitment hysteresis (*Binder et al., 2020*). In this study, we examined the behaviour of pairs of MUs at very low forces, that is, less than 10% MVC. These low-threshold MUs active at such force levels may be more prone to the effect

of sustained PICs (*Heckman et al., 2008b*), which might explain the hysteresis between recruitment and de-recruitment threshold observed for most MUs examined in this experiment.

An inhibitory input is needed to extinguish the impact of PICs on the MU discharge behaviour (*Heckmann et al., 2005*). Such an inhibitory signal leads to the reversal of MU activity to the initial state, once an MU stops firing for a prolonged amount of time. This explains why MU recruitment was not critically altered even after extensive single MU control during this experiment. The very rare cases in which an activation of the higher threshold MU before the lower threshold one occurred may be explained by an incomplete extinction of the PIC effect since in the vast majority of these cases (>90%) the preceding trial required an activation of MU2.

PICs act diffuse across the entire MU pool (*Johnson and Heckman, 2014*). Therefore, the excitability of MU1 and MU2 should be affected similarly by this neuromodulatory mechanism. However, it may be possible that the broad influence of PICs is focalised by local inhibition, which allows suppression of the PIC effect on selective MU pools and sub-pools (*Heckman et al., 2008a*). Inhibitory signals to motor neurons can be facilitated via various mechanisms, including descending pathways, spinal circuitries, or pre-synaptic inhibition (*Heckman et al., 2009*). One intensively studied form of inhibition directly influencing PICs is reciprocal inhibition (*Johnson and Heckman, 2014*; *Heckman et al., 2008a*; *Hultborn et al., 2004*), triggered by changes in the length of antagonist muscles, which can occur, for example, by moderately altering joint angles (*Hyngstrom et al., 2007*). Further, descending inputs may additionally tune interneurons mediating reciprocal inhibition and thus also influence PICs and the excitability of single motor neurons (*Jankowska, 1992*). Synaptic inhibition received by motor neurons follows the size principle. Hence, smaller motor neurons tend to show a greater hyperpolarisation than larger ones for the same inhibitory input (*Henneman et al., 1965*; *Heckman and Binder, 1988*). Consequently, a higher threshold MU may continue firing after a smaller threshold MU already became silent. In our experiment, we did not measure PICs nor inhibitory inputs to MUs. Hence, we cannot pinpoint the exact mechanisms explored by subjects to generate the necessary activation patterns to move the cursor towards TIII. However, we consider that it is likely that a mixture of broadly distributed neuromodulatory mechanisms, such as PICs, and locally acting inhibition may have produced the variability in MU activity that was observed in this study. Similar findings on the increasing variability in MU activation by incorporating neuromodulation and inhibition have been shown in a simulation study (*Powers et al., 2012*).

During a progressive increase in force, recruitment depends only on the MU anatomy and the input received. If humans can learn to leverage potential structures in the CNS that allow selective inputs to MUs (*Marshall et al., 2021*), changes in the recruitment order during this initial phase should be expected. However, it is important to underline that a conclusion of flexible control based on changes in MU recruitment cannot be drawn for time intervals that follow an activation of the MU. In these cases, the de-recruitment of an MU at a force level different from the recruitment threshold could be incorrectly interpreted as an alteration of the recruitment order. Presumably, such changes result from the relative intrinsic excitability of the motor neurons which override the sole impact of the received synaptic input on the recruitment order. Therefore, a direct proof of altered MU recruitment as a consequence of independent input to different MUs needs to be provided during the initial activation phase, that is, an MU with higher recruitment threshold activated before an MU with lower threshold without preceding activations. It is also worth mentioning that this proof should further include MUs with sufficiently different recruitment thresholds since synaptic noise may influence the relative recruitment order for MUs of very similar thresholds (*Heckman and Enoka, 2012*).

We did not get results supporting a *general* flexible control of MUs, that is, volitional activation of higher threshold MUs before lower threshold ones at initial recruitment. However, flexible control of individual MUs could still be a framework explaining how subjects were able to reach the different targets in the 2D space. If this was the case then, since control would be achieved only after MUs were recruited, this would imply that flexible control is state-dependent: it can only be achieved in the context of previously contracted muscle fibres. Such state dependency restricts the possible neural strategies that could allow flexible control of MUs. One possible strategy consistent with such state-dependent control of individual MUs could be relying on an input signal to MUs not directly linked to motor function and non-homogeneously distributed among the MU pool. For example, cortical oscillations could meet these criteria if descending projections of this activity to large and small MUs in a pool differed (*Ibáñez et al., 2021*; *Bräcklein et al., 2021*; *Bräcklein et al., 2022*). A different

alternative that may allow for flexible MU control after initial recruitment is volitional modulation of reciprocal inhibition (*Chen et al., 2006*; *Thompson et al., 2013*) presumably via direct descending commands (*Jankowska, 1992*; *Nielsen et al., 1995*). Such augmentation of synaptic inhibition could potentially lead to changes in the excitability of specific MUs. Such inputs to MUs could provide a certain degree of flexibility to control subgroups of MUs in a muscle volitionally. Future studies are needed to test this hypothesis.

Throughout the 14 days of training, subjects were asked to modulate the discharge rate of MU pairs independently to navigate a cursor as quickly as possible into different targets inside a 2D space. The target and angle hit rates indicate that subjects could achieve control over these single MUs. Moreover, the angle hit analysis (e.g. see *Figure 5A*) revealed that during the last 5 days of training, subjects were able to produce the necessary activation pattern in most trials despite different properties of the selected MUs. Although subjects repeated the target-tracking task with different sets of MUs every day, they consistently reported the use of the same control strategies across days. Hence, these findings suggest that subjects learned to establish universal control strategies that allowed for the achievement of the target task for various combinations of MUs. To reach TI or TII, subjects used precise force control to exert either a low-enough force that only the low threshold MU (MU1) turned active (TI) or a force above the recruitment threshold of both MU1 and MU2 (TII). To reach TII, subjects used strategies that corresponded to a physiological activation of the two MUs similar to the initial force ramp.

As could be expected, subjects had difficulties in navigating the cursor towards TI when the difference in recruitment threshold between MU1 and MU2 was small (≤1% MVC; see *Figure 3D*). In these cases, subjects could not selectively activate MU1 while keeping MU2 deactivated and still reach a discharge rate of MU1 high enough to place the cursor near the centre of TI. When the difference in recruitment threshold was larger (≥3% MVC), the discharge rate of MU1 reached the desired value before the onset of MU2. It seems the discharge rate of MU1 saturated (*Fuglevand et al., 1993*) due to its intrinsic properties, as recently discussed in *Fuglevand et al., 2015*. Unlike for TI and TII, TIII required a more cumbersome approach. To reach TIII, subjects mainly mimicked the trajectory of TII first, that is, activating both MUs, followed by the second stage of control in which the force level was reduced until the lower threshold MU turned off by leveraging the mismatch in the de-recruitment thresholds. In order to then place and hold the cursor inside TIII, subjects gradually increased the force again without reactivating MU1. These second and third stages of control were possible in principle by maintaining a common ionotropic input to the MU pair combined with neuromodulatory input and synaptic inhibition, as described above, even when MU2 was initially de-recruited before MU1. If a direct activation of MU2 would have been possible as it was for MU1, subjects would have chosen to mimic a cursor trajectory along the direct path from the origin to the target centre, as observed for both TI and TII (see *Figure 4*). This almost never occurred in the hundreds of trials tested. Therefore, the sole activation of a higher threshold MU was only possible by exploiting the history-dependent activation of MUs, that is, exclusive firing of MU2 follows the combined activation of MU1 and MU2. The results suggest that this three-stage approach to achieve a hit in TIII as quickly as possible was feasible for the subjects while a more efficient strategy of directly activating MU2 without a preceding activation of MU1 was not.

It has been previously shown that subjects can learn to control MUs independently when exposed to biofeedback on the discharge behaviour (*Formento et al., 2021*; *Basmajian, 1963*). In these previous investigations, the subjects were allowed for contractions along multiple directions. Such variations in force directions (*ter Haar Romeny et al., 1984*; *Desnedt and Gidaux, 1981*) but also other motor behavioural changes, including alternations in postures (*Nardone et al., 1989*), and contraction speed (*Desmedt and Godaux, 1977a*), potentially have an impact on the recruitment order. Similar changes in an MU pool's discharge activity imposed by such behavioural changes were recently suggested in non-human primates (*Marshall et al., 2021*). Therefore, individual MU control may be triggered by small compensatory movements rather than being the result of a dedicated and volitionally controllable individual synaptic inputs. Indeed, independent control of individual MUs would imply that an MU can be controlled independently *of all other MUs*. The fact that a pair of MUs can be controlled independently when varying the task does not imply that the two MUs are independently controlled in absolute terms. They are simply independently controlled with respect to each other. For example, in some tasks or in some conditions, they may be part of different groups of MUs receiving two different common inputs (*Tanzarella et al., 2021*).

A recent study in humans provided evidence for the existence of MU pool synergies similar to the functional grouping of muscles involved in a single movement (*Tanzarella et al., 2021*). The CNS may send a common input to these MU pool synergies, which are not per se limited to innervating only a single muscle (*Laine et al., 2015*). In our experiment, we chose a simple case of an MU pool constituting a functional group during ankle dorsiflexion, that is, MUs innervating the TA. During more complex tasks, for example, movement along multiple directions, the CNS would send different common inputs to certain numbers of groups of MUs. While the synergistic organisation of MUs might be flexible across tasks, for example, movement along multiple directions, it remains yet to be explored if the input to a single functional MU group can be changed volitionally from common to individual while the performed task is maintained. Hence, it is crucial that initial conditions during MU recruitment, such as posture, contraction speed, and force direction, are kept constant when flexible MU recruitment is investigated. MU activation based on changes in behaviour, or the performed motor task, does not indicate that subjects can volitionally trigger MU activity by a selective synaptic input as it is possible for their cortical counterparts. Furthermore, these constraints need to be considered in neural-machine interface applications relying on flexible MU control. Possible extracted control signals may depend on behavioural changes and not on a designated descending control command, and therefore, may not satisfy the intended application. Similar constraints effectively apply for augmentation when the aim is to extend the degrees of freedom that a human can volitional control, that is, adding supernumerary degrees of freedom to the natural ones (*Eden et al., 2022*; *Dominijanni et al., 2021*). In such cases, if the control of a supernumerary degree of freedom is based on single MU activation, this activation must be uncoupled from motor behaviour to ensure coordination between natural and supernumerary effectors. This would correspond to breaking common input into multiple inputs. Nevertheless, even based on behavioural changes, single MU control can be a resource for specific neural-interface applications, for example, in the absence of any additional motor information or to augment a specific motor task (see *Eden et al., 2022* for *augmentation by transfer*).

Previous studies investigating flexible MU control focused on upper-limb muscles (*Formento et al., 2021*; *Basmajian, 1963*; *Marshall et al., 2021*). Here, we extracted single MU activity from the TA since this muscle has properties (e.g. muscle fibres arrangement, proximity to the skin, distribution of innervation zones) that facilitate a reliable decomposition of MU activity using surface recordings (*Barsakcioglu et al., 2021*; *Bräcklein et al., 2021*; *Bräcklein et al., 2022*; *Negro et al., 2016a*; *Dideriksen et al., 2018*; *Del Vecchio et al., 2020*) and a relatively large number of monosynaptic connections with the motor cortex that could potentially be leveraged for direct independent MU control (*Ibáñez et al., 2021*; *Dideriksen et al., 2018*). Our results strongly suggest that subjects do not tend to find or opt for a control strategy that relies on flexible MU recruitment order under constraint isometric conditions. In contrast, the established control strategy seemed to be based on a common synaptic input to the MU pair combined with intrinsic changes of MU excitability due to neuromodulation and inhibition. Although the complexity of tasks in which humans use upper and lower limbs may differ, we are not aware of any evidence suggesting that the CNS employs fundamentally different strategies to orchestrate MU activity in a functional motor neuron pool differently across limbs. Future research on this topic may help to better understand if the CNS may allow for variability in upper-limb MU pools.

Throughout the experiment, subjects were restricted by an ankle dynamometer and instructed to perform dorsiflexion of the ankle only. We measured force due to dorsiflexion but not, for example, rotational forces due to ankle supination. Modest rotations of the ankle can increase the effect of inhibition in specific MUs and thus might affect their excitability (*Hyngstrom et al., 2007*). Such effects were only observed for modest changes in joint angles (~20°) (*Hyngstrom et al., 2007*), which were not possible to perform when the dynamometer restricted the foot. It may be that even very subtle rotations or slight movements of other body parts, such as the knee or hip, may have influenced the discharge activity of specific MUs (*Jankowska, 1992*), potentially a helpful approach for achieving TIII-instructed trials. However, none of the subjects reported the use of compensatory movements as a control strategy. Furthermore, even if the subjects unknowingly performed very subtle compensatory movements in TIII-instructed trials, they would have discovered and chosen this strategy over directly utilising potential descending pathways triggering independent MU control.

To summarise, we have demonstrated the ability to control up to four MUs from a single muscle using real-time feedback on single MU discharge behaviour. Furthermore, we have shown by operant

conditioning that subjects learn concrete control strategies to recruit and de-recruit several MUs volitionally. These strategies exploit orderly recruitment in agreement with the *Henneman's size principle* and a common input to their motor neurons. Conversely, the observed strategies do not leverage potential pathways that may provide selective inputs to single MUs. It is concluded that converting common input to a (synergistic) pool of motor neurons into independent input to single MUs within the same task seems extremely challenging for the CNS.

## Materials and methods

### Subjects

Seven healthy subjects (two females and five males, age: 27.86 ± 4.06 years [μ ± SD]) were recruited for the study of whom three are authors of this article. Four subjects were naïve to the experimental paradigm, while the remaining three were recently exposed to single MU feedback. Experiments were carried out over 14 days in blocks of 4–5 consecutive days with never more than 2 days of break between blocks. Each experimental session lasted approximately 2 hr. One subject withdrew from the experiment after only 10 sessions due to time constraints. The study was approved by the ethics committee at Imperial College London (reference number: 18IC4685).

### Data acquisition

High-density surface electromyogram (HDsEMG) was acquired from the TA of the dominant leg via a 64-electrode grid (5 columns and 13 rows; gold-coated; 1 mm diameter; 8 mm interelectrode distance; OT Bioelettronica, Torino, Italy). The adhesive electrode grid was placed over the muscle belly aligned to the fibre direction. In addition, EMG from the FL, GL, and GM were recorded throughout the experiment via pairs of wet gel electrodes (20 mm interelectrode distance; Ambu Ltd, St Ives, UK) placed over the muscle belly. All EMG signals were monopolar recorded, amplified via a Quattrocento Amplifier system (OT Bioelettronica), sampled at 2048 Hz, A/D converted to 16 bits, and digitally band-pass filtered (10–500 Hz). The foot of the dominant leg was locked into position to allow dorsiflexion of the ankle only. The force due to ankle dorsiflexion (single degree of freedom) was recorded via a CCT TF-022 force transducer, amplified (OT Bioelettronica), and low-pass filtered at 33 Hz. The communication between the amplifier and the computer was conducted via buffers, that is, data packages of 256 samples corresponding to a signal length of 125 ms. All incoming EMG signals were band-pass filtered between 20 and 500 Hz using a fourth-order Butterworth filter. Bipolar derivations were extracted from the filtered EMG signals obtained from the FL, GL, and GM.

### Experimental paradigm

#### Pre-experimental calibration

Subjects were instructed to perform maximum isometric dorsiflexion of the ankle to estimate the MVC level. The obtained MVC was then set as a reference value for the subsequent experimental session. In a sub-MVC task, subjects were instructed to follow a 4 s ramp trajectory (2.5% MVC/s) followed by a constant phase at 10% MVC of 40 s. In both tasks, visual feedback of the force produced by isometric TA contractions was provided. Based on the EMG of the TA recorded during this sub-MVC task, the separation matrix used by an online decomposition algorithm was generated to extract MU discharge behaviour in real time (see *Barsakcioglu et al., 2021* for further explanation). The decomposition results were visually inspected while subjects were instructed to recruit MUs one after another based on the visual feedback provided.

#### Force feedback task

After initialising the real-time decomposition algorithm, subjects were instructed to follow ramp trajectories consisting of a 10 s incline (1% MVC/s) followed by a 10 s plateau at 10% MVC and a 10 s ramp decline (–1% MVC/s) guided by visual feedback of the force. This ramp trajectory was repeated five times with 5 s rest period in between ramps. Only the first ramp was used for further analysis since repeated activation of MUs within a short time may influence the recruitment order (*Gorassini et al., 2002*). If a subject failed to follow the ramp during the initial ramp (which happened in 20.2% of all cases), the consecutive ramp was chosen as the basis for further processing and analysis. Based on the recorded force and underlying MU activity during the incline phases, the recruitment order

of the decomposed MUs was estimated as suggested in *Del Vecchio et al., 2020*. The onset of MU recruitment was defined as the time when a MU started to discharge action potentials at 5 pps or above. The averaged force values, extracted from a 100 ms window centred around this onset of MU activity during the initial ramp, were used to establish the MU recruitment order by ranking MUs based on their corresponding force values in ascending order. The plateau phases of the ramps were used to estimate the average discharge rate of each MU during 10% MVC. These values were later used during the target task to normalise the discharge rates. The decline phase was used to determine the force values associated with MU de-recruitment. The time point of MU de-recruitment was defined by the last action potential discharged before an MU turned 'silent' for at least 1.5 s. Similar to the calculation of the force level needed for MU recruitment, the force level at MU de-recruitment was estimated by the average force value extracted from a 100 ms window at the offset of MU activity across the initial ramp.

## MU selection

Subjects were provided with visual feedback on the ranked MU activity. In an exploration phase (approximately 10 min), subjects were instructed to recruit MU one by one by gradually increasing the contraction level of the TA until all identified MUs were discharging action potentials. The entire pool was divided into two sub-pools comprising the first and last recruited half of MUs, respectively. One pair of MUs with a similar recruitment threshold from each sub-pool was randomly selected. Hereby, MU pairs were excluded from the selection if subjects could not recruit these two MUs one by one even after the initial exploration phase. The MU recruited first in each pair was labelled as *MU1*, while the MU recruited last as *MU2*. The selected MUs and their decoded spiking activity are exemplarily shown in *Figure 1—figure supplement 1*.

## Target task

In the main experiment, subjects navigated a cursor in a 2D space by modulating the discharge rate of MU1 and MU2. The normalised discharge rate of these two MUs was used to span the manifold with units ranging from 0 to 1 along both axes. This target space included three targets of equal size (radius of 10% of the normalised discharge rate) placed along the axes (TI [1 normalised discharge rate MU1; 0 normalised discharge rate MU2], TIII [0; 1]) and the diagonal (TII [1; 1]). In addition, each target was framed by an angle space comprised of a triangle with one corner in the coordinate origin [0; 0] and two sides to be tangent at the circumcircle of the corresponding target. Towards the coordinate origin, the angle area was cropped by a circle centred at the origin with a radius of 40% of the normalised discharge rate. In order to navigate the cursor inside this angle space, subjects would need to generate the same discharge relationship between MU1 and MU2 as for reaching towards the target area but without necessarily matching the exact discharge rate determined during 10% MVC. For example, to place and hold the cursor inside the angle space of TI, subjects would need to keep MU1 active while MU2 was inactive. However, the discharge rate of MU1 could be different from its discharge rate at 10% MVC required to place the cursor on the centre of TI. This means the cursor could be placed at (0.7; 0), resulting in an angle hit but not a target hit.

The discharge behaviour of MU1 and MU2 was decomposed from the acquired HDsEMG in real time. The obtained discharge rates were averaged over the preceding eight buffers (corresponding to 1 s) and normalised by the average discharge rate at 10% MVC of the respective MUs . The cursor movement was updated every buffer (corresponding to 125 ms) and smoothed over six buffers (corresponding to 750 ms) using a moving average. In total, the moving average on the discharge rates and the cursor position resulted in a weighted average on the discharge behaviour of MU1 and MU2 over an effective window of 1625 ms. However, the visual feedback was updated every 125 ms. Because of the update rate, as soon as the discharge behaviour of the selected MU changed, subjects were aware of the change due to the visual display of MU discharges and the resulting cursor movement. In an initial familiarisation phase, subjects could freely move inside the target space and explore different control strategies. During this period, the gain along each axis was set manually to enable subjects to reach the target areas without overexerting themselves to prevent symptoms of muscle fatigue. On average, across all subjects, the gain was increased to 1.15 ± 0.01 for MU1 and 1.16 ± 0.01 for MU2, respectively.

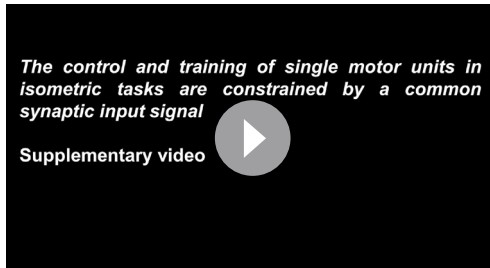

**Video 1.** A simulation of the biofeedback as shown to the subjects. Video illustrates biofeedback environment as presented to the subjects during the target task, showing feedback on the motor unit (MU) spiking activity and the resulting cursor movement in the target space. Simulated data stream based on real data of a TII-instructed trial recorded from one exemplary subject. MU1 is shifted by +250 ms every iteration. Simulated data stream loads buffers of new incoming data every 125 ms as during the experiment. The lower-left window shows the incoming discharge behaviour of the selected MUs MU1 and MU2. The right window shows the targets and cursor movement according to the incoming data stream.

https://elifesciences.org/articles/72871/figures#video1

After familiarisation with the target environment, subjects were asked to rest at the coordinate origin. Once the target-of-interest appeared, indicated in blue, a trial started, and subjects were instructed to navigate the cursor as quickly and as directly as possible into the target area. If the cursor was kept inside the target-of-interest for at least seven consecutive buffers (corresponds to 875 ms; one buffer size longer than the moving average), subjects were granted a target hit, and the trial ended. The trial was also terminated if subjects failed in navigating and holding the cursor inside the target area within 20 s. If the subjects kept the cursor inside the angle area of the target-of-interest for seven consecutive buffers before the trial was terminated, the subjects were granted an angle hit. Therefore, in a single trial, subjects could achieve both an angle and target hit. A target hit was indicated via colour change of the target-of-interest to green, an angle hit to yellow, and a failure (no target nor angle hit within 20 s) to black. Once the trial ended either after 20 s or a target hit, the subject was instructed to navigate the cursor back into the coordinate origin and rest there for at least

2 s. The entire trial cycle is illustrated in *Figure 2*. In total, every target was presented 10 times in randomised order. Moreover, this target task was repeated three times for three different conditions. In condition I, MU1 and MU2 were coming from the lower threshold pair while, in condition II, they were taken from the higher threshold pair. In condition III, MU1 was the lower threshold unit from the low-threshold pair and MU2 the higher threshold MU from the high-threshold pair. After the target task was completed, subjects repeated the force task.

During the familiarisation phase and experiment, subjects received both feedback on the cursor movement and the underlying discharge behaviour of the selected MU pair updated for every buffer of incoming data, that is, 125 ms. *Video 1* visualises a simplified version of the biofeedback environment presented to the subjects.

## Analysis
### Force task
The EMGs of the TA, FL, GL, and GM acquired during both force tasks before and after the target task were rectified and low-pass filtered at 10 Hz with a fourth-order Butterworth filter. Similar to the estimation of the force level at the on- and offset of MU activity , the average global EMG values to MU de-/recruitment of all muscles were calculated. The average value inside a 100 ms window centred around the time point of recruitment and de-recruitment for each MU and muscle was calculated across all ramps. This was separately repeated for all values acquired before and after the target task. Three subjects did not repeat the force task after the target task and only followed a single ramp at the beginning of the experiment. Moreover, no de-recruitment threshold was determined for those subjects.

### Target task
If subjects failed to navigate and hold the cursor for 875 ms inside the target-of-interest within the 20 s time window, the nearest miss was calculated. The nearest miss was defined as the average cursor position over 875 ms with the shortest Euclidean distance towards the centre of the target-of-interest. In addition, as previously described in *Bräcklein et al., 2021*, unintended hits were used as a metric to assess the effectiveness of subjects directly hitting the target-of-interest without unintentionally hitting unselected targets before. Therefore, an unintended hit was classified as the case when

subjects navigated and held the cursor inside an unselected target for at least 875 ms. Unintended hits of the same target could occur multiple times within a single trial if the cursor re-entered the unselected target on several occasions before the trial was terminated. Similarly, unintended angle hits were counted when the cursor was navigated into unselected angle areas, respectively.

TI and TIII required the sole activation of either MU1 or MU2. To assess subjects' performance in navigating the cursor towards these two targets even when neither the target nor the angle area was reached, but, for example, the discharge rate of the target MU was greater than the one of the other, a new performance metric was introduced that compensates shortcomings of traditional metrics, for example, time-to-target and distance-to-target. This new performance metric was designed to endorse scenarios in which this performance metric was defined as

$$performance = \sum_{n=1}^{N} \sqrt{\left(\frac{d(n)}{d_{max}}\right)^2 + \left(\frac{\varphi(n)}{\varphi_{max}}\right)^2},$$

where $d(n)$ is the Euclidean distance between the centres of the cursor and target-of-interest, and $\varphi(n)$ is the angle between the cursor and the target-of-interest at the buffer $n$. $N$ is the total number of buffers recorded in one trial. For TI and TIII, $d_{max}$ was set as the Euclidean distance between the centres of TI and TIII, and $\varphi_{max}$ to 90°. For TII, the target along the diagonal, $d_{max}$, was set to the distance between the centre of TII and the origin, and $\varphi_{max}$ to 45°. When the cursor was held in the origin, that is, no activation of either MU1 or MU2, $\varphi(n)$ was set to $\varphi_{max}$. By summing across all recorded buffers, this metric incorporates the time-to-target and favours those trials in which the cursor was kept close to the target-of-interest even when neither the angle nor the target area was hit. To scale the *performance values* between 0 and 1, the obtained result was normalised with the *worst* and *best performance values* estimated per target. The *best performance value* for each target across subjects was obtained by simulating cursor movement based on artificially generated MU discharge patterns that match the required activation of MU1 and MU2 to hit the respective target. For example, the *best performance* for TI was estimated based on discharge behaviour for MU1 that matched a normalised discharge rate equal to 1 and MU2 equals 0. The performance value during the idealised cursor movement until the target hit was used as the corresponding *best performance value*. For the *worst performance value*, the performance was calculated as if the cursor was kept in the origin for the entire 20 s.

The described metrics were calculated across all conditions and subjects. Three subjects started with condition III only from day 10 onwards.

## Questionnaire

After every condition, subjects were provided with a questionnaire. Subjects were asked to indicate the level of control they had over MU1 when reaching towards TI, over MU2 when going towards TIII, and both MUs when reaching towards TII. Moreover, whether subjects felt they were using a specific strategy when going to the selected target-of-interest and how cognitively demanding it was to control the MUs together and independently was assessed. When applicable, they were asked to explain their strategy. Three subjects did not fill out the questionnaire.

## Validation of online EMG decomposition during recruitment/de-recruitment

A validation study was performed on three subjects (one female and two males, age 32, 24, 25 years) to compare the on- and offset activity of MUs detected via surface and intramuscular EMG recordings. For this purpose, surface EMG signals were recorded in the same way as in the main experimental task by a 64-channel high-density grid placed over the TA muscle and aligned to the fibre direction of the dominant leg. In addition to the non-invasive grid, two fine-wire electrodes were inserted with an insertion angle of 45° to a depth of a few millimetres below fascia, 10 mm proximal with respect to the top of the electrode grid. The foot was locked into position to allow for the ankle dorsiflexion only, as for the main experimental session. All signals were acquired synchronously with a Quattrocento Amplifier system (OT Bioelettronica), sampled at 10,240 Hz, A/D converted to 16 bits, and digitally band-pass filtered (10–4400 Hz).

For the validation test, the participants were provided with visual feedback of the ankle dorsiflexion force and were presented with six target force profiles. All force profiles consisted of a ramp-up

trajectory, a 20 s constant force phase at 10% MVC, and a ramp-down trajectory. The ramp-up and ramp-down phases had a slope of 0.5% MVC/s, 1% MVC/s, 2% MVC/s, 2.5% MVC/s, 5% MVC/s, and 10% MVC/s. Each target was repeated twice. In this way, the contractions included recruitment and de-recruitment for a broad range of contractions speeds.

The HDsEMG signals were real-time decomposed during the contractions in the same way as done in the main experimental session. The intramuscular EMG signals concurrently recorded were analysed offline and decomposed with the EMGLab software (setting a high-pass filtering at 1000 Hz) (*McGill et al., 2005*). The surface EMG online decomposition was compared to the intramuscular EMG decomposition by the rate of agreement, that is, the percentage of discharges identified by both methods within a maximum time difference of 1ms. For this purpose, there was the need to identify the MUs that were commonly identified by intramuscular and surface EMG decomposition. The identification of the pairs of common MUs was performed with a procedure fully independent on the rate of agreement (which was the metrics to be validated). For this purpose, the spike train of each MU identified by intramuscular EMG decomposition was used for spike-triggered averaging the multi-channel surface EMG. This provided the estimate of the action potential waveform shape as detected by the surface EMG grid for the MUs identified by intramuscular EMG decomposition. The spike trains of the MUs identified by the surface EMG decomposition were then used for spike-triggered averaging the surface EMG and therefore to obtain the action potential waveform shapes at the skin surface for the MUs identified by surface EMG decomposition. The matching for identifying common units between the two decomposition processes was then performed by comparing the shapes of the action potentials at the skin surface obtained from the averaging processes based on the spike trains identified by intramuscular and surface EMG decomposition. The comparison was based on the correlation coefficient between action potential waveform shapes, with a threshold of 0.9 set to accept two MUs as commonly detected.

## Statistics

Statistical analysis was conducted using SPSS (IBM, Armonk, NY) and MATLAB (version 2018b, The MathWorks, Inc, Natick, MA) for the linear mixed model analysis. The threshold for statistical significance was set to $p < 0.05$. To avoid accumulation of type I errors, non-parametric tests were used to assess the relationship between variables (*Rochon et al., 2012*). To compare recruitment and de-recruitment thresholds, a linear mixed model with the difference between recruitment and de-recruitment threshold as dependent variable, a fixed effect intercept and a participant specific random intercept was applied using restricted maximum likelihood estimation. Significance of the fixed effect was assessed by an F-test using Satterthwhaite's approximation for the degrees of freedom. For analysing the improvement in target and angle hit rate, performance across and for each target, as well as the relationship between the performance of reaching each target and the difference in recruitment and de-recruitment thresholds between MU2 and MU1, two-sided Wilcoxon signed-rank tests were used. Comparison of mean characteristic forces during indirect hits of TIII and the mean correlation between performance and successive trials was done via Friedman test. A two-sided Wilcoxon signed-rank test was used for post-hoc analysis. The correlation between recruitment and de-recruitment thresholds as well as the within-MU-pair difference in recruitment and de-recruitment thresholds was assessed using Spearman's correlation coefficient.

## Acknowledgements

This study was supported by the EPSRC Centre for Doctoral Training in Neurotechnology and Health and the European Commission grants H2020 NIMA (FETOPEN 899626) and H2020 TRIMANUAL (MSCA 843408).

# Additional information

## Funding

| Funder | Grant reference number | Author |
| --- | --- | --- |
| EPSRC Centre for Doctoral Training in Neurotechnology and Health | | Mario Bräcklein |
| H2020 NIMA | FETOPEN 899626 | Deren Yusuf Barsakcioglu<br>Jonathan Eden<br>Jaime Ibáñez<br>Etienne Burdet<br>Carsten Mehring<br>Dario Farina |
| H2020 TRIMANUAL | MSCA 843408 | Jonathan Eden<br>Etienne Burdet |
| "la Caixa" Foundation | LCF/BQ/PI21/11830018 | Jaime Ibáñez |

The funders had no role in study design, data collection and interpretation, or the decision to submit the work for publication.

## Author contributions

Mario Bräcklein, Conceptualization, Data curation, Formal analysis, Investigation, Methodology, Software, Validation, Visualization, Writing – original draft; Deren Yusuf Barsakcioglu, Conceptualization, Data curation, Formal analysis, Investigation, Methodology, Software, Validation, Visualization, Writing – review and editing; Jaime Ibáñez, Jonathan Eden, Conceptualization, Methodology, Writing – review and editing; Etienne Burdet, Carsten Mehring, Conceptualization, Funding acquisition, Methodology, Writing – review and editing; Dario Farina, Conceptualization, Funding acquisition, Methodology, Project administration, Resources, Supervision, Writing – review and editing

## Author ORCIDs

Mario Bräcklein http://orcid.org/0000-0003-1537-7495
Jaime Ibáñez http://orcid.org/0000-0001-8439-151X
Carsten Mehring http://orcid.org/0000-0001-8125-5205
Dario Farina http://orcid.org/0000-0002-7883-2697

## Ethics

Human subjects: Informed consent and consent to publish was obtained from all subjects. The study was approved by the ethics committee at Imperial College London (reference number: 18IC4685).

## Decision letter and Author response

Decision letter https://doi.org/10.7554/eLife.72871.sa1
Author response https://doi.org/10.7554/eLife.72871.sa2

# Additional files

## Supplementary files

• Transparent reporting form

## Data availability

All data generated or analysed during this study are included in the manuscript.

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
