## [Editor Report]

The present study indicates that humans cannot easily learn to control multiple motor units innervating a single muscle independently. These results suggest that common drive to motor units and the size-recruitment principle impose strong constraints on the motor system and, as such, on the use of high-resolution muscle recordings as a means of controlling brain-machine interfaces.

---

## [Decision Letter]

**Decision letter after peer review:**

Thank you for submitting your article "The control and training of single motor units in isometric tasks are constrained by a common synaptic input signal" for consideration by *eLife*. Your article has been reviewed by 4 peer reviewers, and the evaluation has been overseen by Andrew Pruszynski as the Reviewing Editor and Tirin Moore as the Senior Editor. The following individual involved in review of your submission has agreed to reveal their identity: Andrew Fuglevand (Reviewer #1).

Essential revisions:

1) The paper needs to provide clear demonstration/analysis of the spike sorting algorithm specifically showing that it can accurately differentiate the onset and offset times of pairs motor units selected from Group 1 and pairs selected from Group 2. This can be done based on the authors already-published simultaneous surface/intramuscular recordings but it should also include an analysis/estimate of accuracy when the method is applied to the data in the present paper.

2) Given the smoothing of firing rates being used, the authors need to demonstrate that the participants could have, in principle, used the visual feedback to discriminate recruitment times of motor unit pairs. One way of doing this is generating a series of videos showing the visual feedback that would have been shown to a participant in which two motor units are recruited at different times relative to one another (this would be synthetic data). That is, artificially shift the timing of an MU1 spike train to the right for a Condition III trial - a few videos with shift increments of ~250 ms seem reasonable for such an illustration.

3) The authors suggest PICs as the explanation of selective de-recruitment of lower threshold units. As described by Reviewer #2, inhibitory inputs will also be shaped by the size principle such that low threshold (i.e. high input resistance) neurons will exhibit greater hyperpolarization. This could lead to a situations where lower threshold neurons become deactivated before higher threshold ones. Please consider this and other explanations in the revised Discussion.

4) The authors should directly discuss the differences between upper and lower limbs in terms of control and thus potential deviations the size-recruitment principle.

*Reviewer #1 (Recommendations for the authors):*

1. (pg 2, ln 14) "These results suggest that flexible MU control based on independent synaptic inputs to single MUs is not a simple to learn control strategy" The last phrase "is not a simple to learn control strategy" seems pretty waffly. I would suggest replacing it with something like "is unlikely".

2. (pg 3, ln 26) "and faster" perhaps substitute "and with higher rates".

3. In his Handbook of Physiology Chapter (Henneman & Mendell (1981) Functional organization of motoneuron pool and its inputs) Henneman describes attempts to alter recruitment order with biofeedback that would seem relevant to the present manuscript:

"In six of the nine subjects no changes in recruitment order were observed despite two hours of training and the help of audiovisual feedback. In each experiment recordings were made from many sites, and the subject was encouraged to explore maneuvers that might lead to alteration in recruitment. At each new site at least 20-30 minutes was spent attempting to alter the normal order. In not a single instance, out of hundreds of trials, was anyone of these six subjects able to recruit two units in their usual small-to-Iarge order and then turn off unit 1 without silencing unit 2".

"The results at almost all recording sites in the three remaining subjects were similar to those just described. In each of these subjects, however, there was one site at which some variability in recruitment order was observed. Although one unit was recruited first and dropped out last in the majority of tests, the unit that was usually recruited second was occasionally the first to respond and could then be activated repetitively for some seconds without any activity in the first unit. These changes in recruitment order seemed to occur randomly. None of the subjects could, on demand, activate unit 2 at will or alternate the activity of the two units in sequence."

4. (pg, ln 29) "and appears to remain robust in various scenarios [21], [22]". There would seem to be other citations, perhaps even more relevant than [21],[22], that might be cited here. These include:

• Desmedt JE & Godaux E (1977). Ballistic contractions in man: characteristic recruitment pattern of single motor units of the tibialis anterior muscle. The Journal of Physiology 264, 673-693.

• Thomas JS, Schmidt EM & Hambrecht FT (1978). Facility of motor unit control during tasks defined directly in terms of unit behaviors. Experimental Neurology 59, 384-397

• Thomas CK, Ross BH & Stein RB (1986). Motor-unit recruitment in human first dorsal interosseous muscle for static contractions in three different directions. Journal of Neurophysiology 55, 1017-1029

• Thomas CK, Ross BH & Calancie B (1987). Human motor-unit recruitment during isometric contractions and repeated dynamic movements. Journal of Neurophysiology 57, 311-324.

• Jones KE, Lyons M, Bawa P & Lemon RN (1994). Recruitment order of motoneurons during functional tasks. Exp Brain Res 100, 503-508

5. (pg 14, ln 5) "This indicates that subjects experienced difficulties in keeping MU2 active while MU1 is inactive in order to reach TIII when their difference in recruitment threshold was large". "Large" is a relative term. Indeed, the actual difference in recruitment thresholds was quite small, on the order of only 6 – 10 % MVC. Perhaps instead state something like "when their difference in recruitment threshold was relatively large (6 – 10 % MVC)."

6. (pg 10 ln 27 ) [This is a minor point and needs to be addressed only if the authors wish to] "in 64.73% a selective MU was de-recruited at a force level below its initial recruitment threshold". One likely explanation for this is that during the decrease in force phase, subjects slightly increased activity of the antagonist muscles (De Luca CJ & Mambrito B (1987). Voluntary control of motor units in human antagonist muscles: coactivation and reciprocal activation. J Neurophysiol 58, 525-542). Even a modest degree of antagonist activity would cause the net (measured) force at derecruitment of a MU to be somewhat less, even though the muscle (TA) force might still be the same as at recruitment (Patten C & Kamen G (2000). Adaptations in motor unit discharge activity with force control training in young and older human adults. Eur J Appl Physiol 83, 128-143; Fuglevand AJ, Dutoit AP, Johns RK & Keen DA (2006). Evaluation of plateau-potential-mediated "warm up" in human motor units. The Journal of Physiology 571, 683-693)

7. (pg 19 ln 4) "An inhibitory input is needed to extinguish the impact of PICs on the MU discharge behaviour." A citation should probably be included here.

8. (pg 21, ln 1) "but also other motor behavioural changes, including alternations in postures [36], and contraction speed [15], are well known factors that impact the recruitment order." However, plenty of other studies suggest that these factors have little clear-cut effect on recruitment order (see citations under point 4 above).

9. (pg 21, ln 3) "changes in a MU pool's discharge activity imposed by such behavioural changes were recently confirmed". The word "confirmed" is far too strong. The paper cited has not undergone peer review. Moreover, the results of that manuscript are questionable for several technical reasons. Suggest change "confirmed" to "suggested".

10. (pg 21, ln 10) Paragraph beginning "A recent study in humans provided evidence for the existence of MU pool synergies" seems somewhat ancillary to the main topic of this paper and could be eliminated.

*Reviewer #2 (Recommendations for the authors):*

1. Because the findings depend on the reliable isolation of single MUs, the authors should provide additional characterization of this process. (A) Please show a segment of raw data (as in Figure 1, lower middle inset) with the spike times for selected MUs highlighted. (B) Provide more details on quality metrics (such as the rate of agreement with manually-curated offline decomposition, as in ref. 25) or criteria (even if these are qualitative). A sensitivity analysis to determine whether the main results are robust to more stringent isolation criteria might also be useful.

2. Ideally, readers should be able to interpret the figures with minimal reference to the text or caption. To this end, additional annotation in Figure 3A and 4 indicating that the y-axis corresponds to the higher-threshold unit would be useful. Similarly, in Figure 5A, please provide additional annotation for Conditions I-III (e.g., "low-low," "high-high," and "low-high"), as well as labels for the Y axes of the confusion matrices (e.g., "intended target" or similar). I would also recommend a uniform color scheme for the entries in the confusion matrices in Figure 5A (e.g., a monochromatic scale with limits of 0-200%), to allow an easy graphical comparison across rows, columns, and conditions.

3. Please show a scatterplot with the TIII hit rates on TIII-instructed trials vs threshold difference for all pairs.

4. A more explicit description of the decoding scheme would help. It appears that x-y cursor positions are weighted averages of MU1 / MU2 discharge rates over the previous second; this could be stated more directly, and additional filtering procedures, if any, described.

5. Figure 1: the "green arrows" indicating the electrodes in the caption do not appear to be in the panel.

*Reviewer #3 (Recommendations for the authors):*

None

*Reviewer #4 (Recommendations for the authors):*

1. Since papers in *eLife* are aimed toward a broad readership, the writing could be clearer at certain points. For example:

a. The acronym DR is never spelled out.

b. The term buffer is confusing.

c. Some statements were a bit vague. For example, page 3, line 12-14; page 7, line 18-19; page 21, line 27-28

[Editors' note: further revisions were suggested prior to acceptance, as described below.]

Thank you for resubmitting your work entitled "The control and training of single motor units in isometric tasks are constrained by a common synaptic input signal" for further consideration by *eLife*. Your revised article has been evaluated by Tirin Moore (Senior Editor) and Andrew Pruszynski (Reviewing Editor).

The manuscript has been substantially improved but there are some remaining issues that need to be addressed, as described below:

1. There remains some concern about the validation (required revision #1 in the previous review). First, it would be good to show in a scatter plot the agreement fraction during recruitment vs. derecruitment as calculated for the common MUs (instead of reporting the average epoch-specific fraction across all MUs). Such a plot could highlight any consistent bias between the two epochs. Second, it is unclear how you decide on common MUs. Is this decision related in some way to the agreement fraction? If so, this appears to be a bit of a circular argument. Please explain why it is not circular in the manuscript.

2. The title needs to be modified to accurately reflect the main findings of the study. Specifically, please remove the word synaptic as the precise mechanism is ultimately unknown in recruitment and de-recruitment. One possibility might be: "The recruitment of single motor units in a trained isometric task is constrained by a common input signal"

3. Similar to above, the end of the abstract should also be modified to ensure it accurately reflects the bounds of the paper. The last three sentences are presently:

"These strategies rarely corresponded to a volitional control of independent input signals to individual MUs. Conversely, MU activation was consistent with a common input to the MU pair, while individual activation of the MUs in the pair was predominantly achieved by alterations in de-recruitment order that could be explained with history-dependent changes in motor neuron excitability. These results suggest that flexible MU control based on independent synaptic inputs to single MUs is unlikely."

Suggestion to clarify: First sentence: "… independent input signals in individual MUs during recruitment/onset of muscle activity". Last sentence: "These results suggest that flexible MU recruitment based on independent synaptic inputs to single MUs is unlikely, although de-recruitment might reflect either varying inputs or PIC"

---

## [Author Response]

Essential revisions:1) The paper needs to provide clear demonstration/analysis of the spike sorting algorithm specifically showing that it can accurately differentiate the onset and offset times of pairs motor units selected from Group 1 and pairs selected from Group 2. This can be done based on the authors already-published simultaneous surface/intramuscular recordings but it should also include an analysis/estimate of accuracy when the method is applied to the data in the present paper.

We have decided to perform a new experiment specifically for testing accuracy in the new conditions of this study. As indicated above, the developed algorithm for online decomposition has been extensively validated in stable force contractions without recruitment/de-recruitment. We have therefore added new concurrent intramuscular and surface EMG data in conditions of recruitment/derecruitment and have looked at the rate of agreement between the decompositions from the two types of signals. We underline here that this is the best possible validation for a surface EMG decomposition algorithm, as we discussed previously. The new experiment and results are described in the revised manuscript (please see p. 29-30 Section 4.6 Validation of online EMG decomposition during recruitment/de-recruitment; Section 2.2 in Results on p. 15-17). Briefly, the new experimental protocol was performed on three subjects and included online surface EMG decomposition in the same way as done in the main experimental session. In these new measures, however, intramuscular signals (wire electrodes) were concurrently recorded. Since the focus was to prove that the online decomposition could track recruitment and de-recruitment with high accuracy, the subjects were asked to follow force profiles of increasing and decreasing force at six different speeds. This allowed to look at recruitment/de-recruitment in a variety of conditions, from slow to fast (the slopes were 0.5%MVC/s, 1%MVC/s, 2%MVC/s, 2.5%MVC/s, 5%MVC/s, and 10%MVC/s). Each force trajectory and force speed was repeated twice. Thus, in total we recorded data from 36 contractions (3 subjects, 6 force speeds, two contractions for each speed), with concurrent data from surface and intramuscular electrodes. The surface signals were decomposed online during the contractions, in the same way as in the main experimental session, and the decomposition result was stored for further analysis. The intramuscular signals were decomposed offline by a semi-automatic procedure, with a validated and extensively used intramuscular EMG decomposition algorithm (which is also freely available; see Methods). The two decompositions were assessed with the rate of agreement, that is the percentage of discharge instants detected by both decompositions within a 1-ms delay margin. This is the same metrics for validation we previously used for analysing stable force contractions and it is a conservative metrics of accuracy (meaning that the actual accuracy in surface decomposition is not inferior and likely superior to the one estimated by this metrics). The results were very clear and indicated (1) that the overall accuracy was comparable to those observed for stable force contractions in previous validation studies (>90% in all cases), and (2) that the errors were distributed uniformly during the contraction, without any preferential distribution around the time of recruitment of derecruitment. Notably, we never observed an error in the first discharge (onset) or the last discharge (offset), meaning that the recruitment and derecruitment of motor units was detected by the online decomposition with an accuracy of 1ms (margin used in the time alignment between discharge times detected by the surface and intramuscular decomposition). These results strongly support the conclusion that the online surface EMG decomposition used in this study during variable force contractions with recruitment and de-recruitment of motor units achieved similar (very highly accurate) performance as the decomposition in static contractions. We are particularly satisfied of this added validation since it can serve as a reference for future studies performed in similar conditions.

The revised manuscript includes all the above information in mode details, as well as a representative figure of the validation procedure and numerical results.

2) Given the smoothing of firing rates being used, the authors need to demonstrate that the participants could have, in principle, used the visual feedback to discriminate recruitment times of motor unit pairs. One way of doing this is generating a series of videos showing the visual feedback that would have been shown to a participant in which two motor units are recruited at different times relative to one another (this would be synthetic data). That is, artificially shift the timing of an MU1 spike train to the right for a Condition III trial -- a few videos with shift increments of ~250 ms seem reasonable for such an illustration.

We have added a video (Video 1; p. 31, l. 8-14) showing the biofeedback environment presented to the subjects with real data from a TII-instructed trial. The trial was repeated nine times, with MU1 being artificially delayed each iteration by 250ms. The video layout shows the feedback subjects have received, i.e. cursor movement and the underlying neural activity of the selected MU pair. Indication of current force level, target-of-interest, angle or target hits were neglected in this simulation. In addition, we clarified throughout the manuscript that subjects were provided with realtime feedback on both the discharge activity of the selected MU pair and the resulting cursor movement.

3) The authors suggest PICs as the explanation of selective de-recruitment of lower threshold units. As described by Reviewer #2, inhibitory inputs will also be shaped by the size principle such that low threshold (i.e. high input resistance) neurons will exhibit greater hyperpolarization. This could lead to a situations where lower threshold neurons become deactivated before higher threshold ones. Please consider this and other explanations in the revised Discussion.

We agree with the reviewer’s comment and thus updated the Discussion accordingly.

4) The authors should directly discuss the differences between upper and lower limbs in terms of control and thus potential deviations the size-recruitment principle.

Following the reviewer's comment, we have included a new paragraph addressing the point about potential differences between upper- and lower-limbs and justifying the use of a lower leg muscle in our work at p. 22, l. 16-29.

Reviewer #1 (Recommendations for the authors):1. (pg 2, ln 14) "These results suggest that flexible MU control based on independent synaptic inputs to single MUs is not a simple to learn control strategy" The last phrase "is not a simple to learn control strategy" seems pretty waffly. I would suggest replacing it with something like "is unlikely".

We changed the last phrase of the abstract accordingly.

2. (pg 3, ln 26) "and faster" perhaps substitute "and with higher rates".

Thank you for this comment; we have adapted the manuscript accordingly.

3. In his Handbook of Physiology Chapter (Henneman & Mendell (1981) Functional organization of motoneuron pool and its inputs) Henneman describes attempts to alter recruitment order with biofeedback that would seem relevant to the present manuscript:"In six of the nine subjects no changes in recruitment order were observed despite two hours of training and the help of audiovisual feedback. In each experiment recordings were made from many sites, and the subject was encouraged to explore maneuvers that might lead to alteration in recruitment. At each new site at least 20-30 minutes was spent attempting to alter the normal order. In not a single instance, out of hundreds of trials, was anyone of these six subjects able to recruit two units in their usual small-to-Iarge order and then turn off unit 1 without silencing unit 2"."The results at almost all recording sites in the three remaining subjects were similar to those just described. In each of these subjects, however, there was one site at which some variability in recruitment order was observed. Although one unit was recruited fIrst and dropped out last in the majority of tests, the unit that was usually recruited second was occasionally the fIrst to respond and could then be activated repetitively for some seconds without any activity in the first unit. These changes in recruitment order seemed to occur randomly. None of the subjects could, on demand, activate unit 2 at will or alternate the activity of the two units in sequence."

Thank you for pointing this out. We have included the reference at p. 12, l. 14-15.

4. (pg, ln 29) "and appears to remain robust in various scenarios [21], [22]". There would seem to be other citations, perhaps even more relevant than [21],[22], that might be cited here. These include:• Desmedt JE & Godaux E (1977). Ballistic contractions in man: characteristic recruitment pattern of single motor units of the tibialis anterior muscle. The Journal of Physiology 264, 673-693.• Thomas JS, Schmidt EM & Hambrecht FT (1978). Facility of motor unit control during tasks defined directly in terms of unit behaviors. Experimental Neurology 59, 384-397• Thomas CK, Ross BH & Stein RB (1986). Motor-unit recruitment in human first dorsal interosseous muscle for static contractions in three different directions. Journal of Neurophysiology 55, 1017-1029• Thomas CK, Ross BH & Calancie B (1987). Human motor-unit recruitment during isometric contractions and repeated dynamic movements. Journal of Neurophysiology 57, 311-324.• Jones KE, Lyons M, Bawa P & Lemon RN (1994). Recruitment order of motoneurons during functional tasks. Exp Brain Res 100, 503-508

We agree that these references add further substance to this claim and thus added them to the manuscript.

5. (pg 14, ln 5) "This indicates that subjects experienced difficulties in keeping MU2 active while MU1 is inactive in order to reach TIII when their difference in recruitment threshold was large". "Large" is a relative term. Indeed, the actual difference in recruitment thresholds was quite small, on the order of only 6 – 10 % MVC. Perhaps instead state something like "when their difference in recruitment threshold was relatively large (6 – 10 % MVC)."

We followed the reviewer’s suggestion.

6. (pg 10 ln 27 ) [This is a minor point and needs to be addressed only if the authors wish to] "in 64.73% a selective MU was de-recruited at a force level below its initial recruitment threshold". One likely explanation for this is that during the decrease in force phase, subjects slightly increased activity of the antagonist muscles (De Luca CJ & Mambrito B (1987). Voluntary control of motor units in human antagonist muscles: coactivation and reciprocal activation. J Neurophysiol 58, 525-542). Even a modest degree of antagonist activity would cause the net (measured) force at derecruitment of a MU to be somewhat less, even though the muscle (TA) force might still be the same as at recruitment (Patten C & Kamen G (2000). Adaptations in motor unit discharge activity with force control training in young and older human adults. Eur J Appl Physiol 83, 128-143; Fuglevand AJ, Dutoit AP, Johns RK & Keen DA (2006). Evaluation of plateau-potential-mediated "warm up" in human motor units. The Journal of Physiology 571, 683-693)

We have added this concept to the Discussion at p. 18, l. 9-11.

7. (pg 19 ln 4) "An inhibitory input is needed to extinguish the impact of PICs on the MU discharge behaviour." A citation should probably be included here.

We have added reference [31] to this sentence.

8. (pg 21, ln 1) "but also other motor behavioural changes, including alternations in postures [36], and contraction speed [15], are well known factors that impact the recruitment order." However, plenty of other studies suggest that these factors have little clear-cut effect on recruitment order (see citations under point 4 above).

We have toned down our claim at p. 21, l. 15.

9. (pg 21, ln 3) "changes in a MU pool's discharge activity imposed by such behavioural changes were recently confirmed". The word "confirmed" is far too strong. The paper cited has not undergone peer review. Moreover, the results of that manuscript are questionable for several technical reasons. Suggest change "confirmed" to "suggested".

We have modified the manuscript accordingly.

10. (pg 21, ln 10) Paragraph beginning "A recent study in humans provided evidence for the existence of MU pool synergies" seems somewhat ancillary to the main topic of this paper and could be eliminated.

This paragraph helps clarifying the importance to constrain the functional output of a single anatomical unit, i.e. a muscle, when flexible MU control based on individual descending commands is studied. Recent investigations have shown that a single anatomical unit can receive multiple common inputs to sub-pools of MUs. Thus, flexible MU control within this single anatomical unit can be triggered by changing the motor task the muscle is involved in. Therefore, this paragraph may help the consideration and design of future studies aiming to investigate flexible MU control due to selective descending inputs.

Reviewer #2 (Recommendations for the authors):1. Because the findings depend on the reliable isolation of single MUs, the authors should provide additional characterization of this process. (A) Please show a segment of raw data (as in Figure 1, lower middle inset) with the spike times for selected MUs highlighted. (B) Provide more details on quality metrics (such as the rate of agreement with manually-curated offline decomposition, as in ref. 25) or criteria (even if these are qualitative). A sensitivity analysis to determine whether the main results are robust to more stringent isolation criteria might also be useful.

We have added Figure 1 – —figure supplement 1 (p. 31, l. 2-7) to address the reviewer’s first request. Moreover, we have added a very careful validation of the decomposition with new measures done with concomitant surface and intramuscular recordings during recruitment/de-recruitment of motor units at variable recruitment speed. The details of these new measures and analyses are specified in the reply to the first request for essential changes (see above).

2. Ideally, readers should be able to interpret the figures with minimal reference to the text or caption. To this end, additional annotation in Figure 3A and 4 indicating that the y-axis corresponds to the higher-threshold unit would be useful. Similarly, in Figure 5A, please provide additional annotation for Conditions I-III (e.g., "low-low," "high-high," and "low-high"), as well as labels for the Y axes of the confusion matrices (e.g., "intended target" or similar). I would also recommend a uniform color scheme for the entries in the confusion matrices in Figure 5A (e.g., a monochromatic scale with limits of 0-200%), to allow an easy graphical comparison across rows, columns, and conditions.

We have changed all figures following the reviewer’s suggestion.

3. Please show a scatterplot with the TIII hit rates on TIII-instructed trials vs threshold difference for all pairs.

The reviewer’s suggestion is now included in Figure 3 – —figure supplement 1B (p. 32, l. 112).

4. A more explicit description of the decoding scheme would help. It appears that x-y cursor positions are weighted averages of MU1 / MU2 discharge rates over the previous second; this could be stated more directly, and additional filtering procedures, if any, described.

We have changed the description of the weighted average in the manuscript at p. 26, l. 21-28.

In addition, subjects received feedback on the discharge behaviour of the selected MU pair and the corresponding cursor movement. We changed the manuscript at p. 4, l. 6 accordingly to clarify the biofeedback modalities used in the experiment and at p. 27, l. 17-20. To further clarify the decoding of incoming MU1/MU2 discharge rates into cursor movement, we included Video 1 (p. 31, l. 8-14; please see the attached video file).

Reviewer #4 (Recommendations for the authors):1. Since papers in eLife are aimed toward a broad readership, the writing could be clearer at certain points. For example:a. The acronym DR is never spelled out.

We spelt out the acronym DR (discharge rate) throughout the manuscript.

We have elaborated and clarified all statements.

[Editors' note: further revisions were suggested prior to acceptance, as described below.]

The manuscript has been substantially improved but there are some remaining issues that need to be addressed, as described below:1. There remains some concern about the validation (required revision #1 in the previous review). First, it would be good to show in a scatter plot the agreement fraction during recruitment vs. derecruitment as calculated for the common MUs (instead of reporting the average epoch-specific fraction across all MUs). Such a plot could highlight any consistent bias between the two epochs. Second, it is unclear how you decide on common MUs. Is this decision related in some way to the agreement fraction? If so, this appears to be a bit of a circular argument. Please explain why it is not circular in the manuscript.

Thank you for further concerns on the validation, which we are happy to address.

For the first point on validation:

We would like to underline the following result reported in the manuscript:

“Notably, we never observed an error in the first discharge (onset) or the last discharge (offset), meaning that the recruitment and de-recruitment times were detected by the online decomposition with an accuracy of 1ms ….”

Thus, the recruitment and derecruitment thresholds (defined as the instants of first and last activation) for all common motor units were identified with a perfect accuracy according to the time margin adopted in the validation (1 ms). A scatter plot of the time instants for recruitment and derecruitment thresholds for all common motor units would thus simply be the identical line with a variability bounded to 1 ms. This would not be informative since the text already report the needed information.

In addition, we already reported the rate of agreement separately for the recruitment phase (ramp up), the stable force phase, and the derecruitment phase (ramp down). Note that the standard deviation for rate of agreement for each of these phases does not exceed 2%. A scatter plot representing the rate of agreement is not possible since when there is no agreement, one spike is missing in one of the two decompositions, and therefore we cannot plot the discharge times in a scatter plot (one of the two coordinates is missing).

We hope the above clarifications provide all the further needed information on the first point raised on validation.

For the second point on validation:

The reviewer is right in that we used the rate of agreement for detecting common motor units. This has been done in previous studies (cited in the manuscript) and consists in putting a relatively low threshold value on the rate of agreement to accept two motor units as commonly detected. In our case, we put a threshold of 30% in rate of agreement. This threshold is so low that does not bias the actual result of validation, as discussed in previous studies. Having said that, in order to fully avoid any doubts on a circular argument, we have now adopted a new criterion for identifying common motor units, fully independent on the rate of agreement. The new implemented procedure is the following. The spike train of each motor unit identified by intramuscular EMG decomposition was used for spiketriggered averaging the multi-channel surface EMG. This provided the estimate of the action potential waveform shape as detected by the surface EMG grid for the motor units identified by intramuscular EMG decomposition. The spike trains of the motor units identified by the surface EMG decomposition were then used for spike-triggered averaging the surface EMG and therefore to obtain the action potential waveform shape at the skin surface for the motor units identified by surface EMG decomposition. The matching for identifying common units was then performed by comparing the shapes of the action potentials at the skin surface obtained from intramuscular and surface EMG decomposition. The comparison was based on a correlation coefficient between action potential waveform shapes >0.9. This approach is fully independent on the rate of agreement. Not surprisingly, when we implemented this approach, we obtained exactly the same 16 common units that we identified by the threshold of 30% on rate of agreement. The results on validation have therefore remained exactly the same. We have now described the new method for selecting common units in the revised manuscript. This new method avoids any doubts on a potential circular reasoning (see p. 30, l. 2-16).

2. The title needs to be modified to accurately reflect the main findings of the study. Specifically, please remove the word synaptic as the precise mechanism is ultimately unknown in recruitment and de-recruitment. One possibility might be: "The recruitment of single motor units in a trained isometric task is constrained by a common input signal"

We have changed the title of the manuscript to “The control and training of single motor units in isometric tasks are constrained by a common input signal” (p.1, l. 1-2).

3. Similar to above, the end of the abstract should also be modified to ensure it accurately reflects the bounds of the paper. The last three sentences are presently:"These strategies rarely corresponded to a volitional control of independent input signals to individual MUs. Conversely, MU activation was consistent with a common input to the MU pair, while individual activation of the MUs in the pair was predominantly achieved by alterations in de-recruitment order that could be explained with history-dependent changes in motor neuron excitability. These results suggest that flexible MU control based on independent synaptic inputs to single MUs is unlikely."Suggestion to clarify: First sentence: "… independent input signals in individual MUs during recruitment/onset of muscle activity". Last sentence: "These results suggest that flexible MU recruitment based on independent synaptic inputs to single MUs is unlikely, although de-recruitment might reflect either varying inputs or PIC"

The last three sentences of the abstract were modified accordingly to suggestion of the editors (p. 2, l. 12 & 15-17):

“These strategies rarely corresponded to a volitional control of independent input signals to individual MUs during the onset of neural activity. Conversely, MU activation was consistent with a common input to the MU pair, while individual activation of the MUs in the pair was predominantly achieved by alterations in de-recruitment order that could be explained with history-dependent changes in motor neuron excitability. These results suggest that flexible MU recruitment based on independent synaptic inputs to single MUs is unlikely, although de-recruitment might reflect varying inputs or modulations in the neuron's intrinsic excitability.”